



# Optimization of the Picarro L2140-i Cavity Ring Down Spectrometer for Routine Measurement of Triple Oxygen Isotope Ratios in Meteoric Waters

Jack A. Hutchings[1], Bronwen L. Konecky[1]

[1]Department of Earth and Planetary Sciences, Washington University in St. Louis, St. Louis, 63130, United States of America

*Correspondence to*: Jack A. Hutchings (jackh@wustl.edu)

**Abstract.** The demanding precision of triple oxygen isotope ($\Delta^{17}O$) measurements in water has restricted their measurement to dual-inlet mass spectrometry until the recent development of commercially available infrared-laser analyzers. Laser-based measurements of triple oxygen isotope ratios are now increasingly performed by laboratories seeking to better constrain the source and history of meteoric waters. However, in practice, these measurements are subject to large analytical errors that remain poorly documented in scientific literature and by instrument manufacturers, which can effectively restrict the confident application of $\Delta^{17}O$ to settings where variations are relatively large (~ 25-60 per meg). We present our operating method of a Picarro L2140-i cavity ringdown spectrometer during the analysis of low-latitude rainwaters where confidently resolving daily variations in $\Delta^{17}O$ (differences of ~10-20 per meg) was desired. Our approach was optimized over ~3 years and uses a combination of published best-practices plus additional steps to combat spectral contamination of trace amounts of dissolved organics, which, for $\Delta^{17}O$, emerges as a much more substantial problem than previously documented, even in pure rainwater. We resolve the extreme sensitivity of the $\Delta^{17}O$ measurement to organics through their removal via Picarro's micro-combustion module, whose performance is evaluated each sequence using alcohol-spiked standards. While correction for sample-to-sample memory and instrumental drift significantly improves traditional isotope metrics, these corrections have only marginal impact (0-1 per meg error reduction) on $\Delta^{17}O$. Our post-processing scheme uses the analyzer's high-resolution data, which improves $\delta^2H$ measurement (0.25 ‰ error reduction) and allows for much more rich troubleshooting and data-processing compared to the default user-facing data output. In addition to competitive performance for traditional isotope metrics, we report a long-term, control standard root-mean-square-error for $\Delta^{17}O$ of 11 per meg. Overall performance ($\Delta^{17}O$ error of 7 per meg, calculated by averaging 3 replicates spread across distinct, independently calibrated sequences) is comparable to mass spectrometry and requires only ~6.3 h per sample. We demonstrate the impact of our approach using a rainfall dataset from Uganda and offer recommendations for other efforts that aim to measure meteoric $\Delta^{17}O$ via CRDS.

## 1 Introduction

The stable isotopic composition of water was among the first applications of isotope ratio mass spectrometry (Dansgaard, 1964; Epstein and Mayeda, 1953) and continues to be a critically useful tool for studying the hydrologic cycle (Bowen et al.,





2019; Gat, 1996). The most common form of water, $^1H_2^{16}O$ is measured as a ratio against its heavier, singly-substituted isotopologues: $^2H^1H^{16}O$, $^1H_2^{17}O$, $^1H_2^{18}O$. Historically, the $^2H:^1H$ and $^{18}O:^{16}O$ variations, reported as $\delta^2H$ and $\delta^{18}O$, respectively, have been the primary targets for isotopic analysis. More recently, $^{17}O:^{16}O$ variations, especially in tandem with $^{18}O:^{16}O$, have found applications as a new secondary measurement complementary to deuterium excess (d-excess = $\delta^2H - 8 \times \delta^{18}O$) capable of tracing a range of processes including atmospheric vapor formation conditions (Uechi and Uemura, 2019), mixing of

differentially evaporated waters (Surma et al., 2018; Voigt et al., 2021), raindrop re-evaporation (Landais et al., 2010), and others (Aron et al., 2021).. The coupled variations of the triple oxygen isotope system, calculated relative to a reference slope and referred to in the literature as $^{17}O$-Excess or $\Delta^{17}O$ (hereafter, $\Delta^{17}O$), are interpreted at the per meg ($10^6$, or parts per million) level rather than the typical per mil ($10^3$, ‰, or parts per thousand). Although multiple formulations exist, throughout this paper we use log-transformation of the primary oxygen isotope ratios and an empirical global reference slope of 0.528

calculates values relative to an empirical global slope (Aron et al., 2021; Luz and Barkan, 2010); see Aron et al. (2021) for a review of reference slope choices.

The precision required to measure $\Delta^{17}O$ within the range of natural variation was first developed using dual-inlet isotope ratio mass spectrometry (DI-IRMS) after conversion of $H_2O$ to $O_2$ (Barkan and Luz, 2005). Later, isotope ratio infrared-laser

spectroscopy (IRIS) instruments were developed to perform triple oxygen isotope measurements without prior conversion of water to other species (Berman et al., 2013; Steig et al., 2014). Compared to DI-IRMS, IRIS instruments cost less (~$100,000 versus $250,000), require less operator expertise, and perform their analyses without any modification of the original sample. Since the inception of IRIS techniques in the 2000s, the primary advantages of DI-IRMS have been improved precision (Wassenaar et al., 2018) and insensitivity to organic contamination (West et al., 2010a).


The Picarro L2140-i is a cavity ringdown IRIS designed to measure the near-infrared absorption of the four previously mentioned isotopologues of water and, thus, the $\Delta^{17}O$ parameter. The L2140-i is distinguished from prior models by the inclusion of a second laser (required for $^{17}O$ analysis) and a laser-current-tuner, which reduces instrument noise and increases the frequency of measurements per second (400-500 ringdowns compared to 200-400 in older models) (Steig et al., 2014). The

instrument operates by producing a laser beam with a specific wavenumber, achieving resonance within the measurement cavity, building light intensity within the cavity under resonant conditions, and then deactivating the laser beam and measuring the decay time of the laser, which is quantitatively linked to absorption at that wavenumber. Ringdowns are performed across the wavenumbers of the target isotopologues to generate a spectrum, and isotopologue peaks are integrated as described in Steig et al. (2014). Integrated absorption (A) values for each isotopologues are then used to calculate isotope ratios (e.g., $^{18}R$

= $A(^1H_2^{18}O) / A(^1H_2^{16}O)$). The instrument readout and user-accessible data present raw (uncalibrated) delta values using these ratios (e.g., $\delta^{18}O$). The L2140-i is, fundamentally, a continuous flow device and can be used as such for monitoring of water vapor (Brady and Hodell, 2021; Steig et al., 2021), although a common application involves coupling to a vaporizer for discrete measurement of water samples (Schauer et al., 2016).

The L2140-*i* has a relatively limited number of user-changeable operational modes. The largest distinction is between 'Normal Mode' and '17O Mode', the former of which only measures $^1H_2^{16}O$, $H^2H^{16}O$, and $^1H_2^{18}O$ while the latter includes $^1H_2^{17}O$. To measure a full spectrum of targeted water isotopologues, 30 discrete spectra are measured in 'Normal Mode' while 52 discrete measurements are made in '17O Mode'. This number of discrete measurements alone should enable higher precision measurements of $\delta^2H$ and $\delta^{18}O$ while in 'Normal Mode' than in '17O Mode' due to the increased dwell time of the instrument

on those spectra and concomitant reduction in noise. When used as a discrete liquid sampler, a large excess of sampling material allows varying the duration of sampling of each injection to either increase sample throughout (lower sample measurement times) or increase precision (longer sample measurement times). These trade-offs are achieving using 'High Throughput' and 'High Precision' modes, consistent with injection-to-injection periods of ~4 and ~9 min, respectively (Picarro Inc., 2015b). Schauer et al. (2016) found that an even longer sampling duration (denoted here as 'Long Pulse') that results in

an injection period of ~14.4 min further improved the measurement precision of oxygen stable isotopes for the L2140-*i*.Lastly, if included, a 'micro-combustion module', or MCM, can be used to remove organic contaminants that may absorb in the same range as water and produce spectral interference (Wassenaar et al., 2018; West et al., 2010b).

Our lab has been operating the L2140-*i* in 'MCM 17O Long Pulse' mode since February of 2019, similar to the sampling

duration of Schauer et al. (2016). Since January 2020 we have operated routinely with the MCM on to remove organic contaminants present in environmental waters, as we have found these to interfere strongly with the $\Delta^{17}O$ analysis (see Sect. 3.3). Unknown samples measured during this period were meteoric waters (predominantly precipitation) collected from various field campaigns in the U.S. and in East Africa.

We present four key research topics during a 2 year measurement period for the operation of the Picarro L2140-*i* : (1) sequence structure and post-processing corrections with a limited dataset to demonstrate their effectiveness; (2) a full-factorial experiment comparing instrument modes ('Normal Mode' versus '17O Mode') and analysis times ('High Precision' versus 'Long Pulse') to assess their effects on short-term precision and accuracy; (3) a demonstration of high sensitivity of the L2140-*i* to organic interference when measuring $\Delta^{17}O$; (4) a report of error metrics for known standards during our operation of the

instrument. Our experience with the L2140-*i* leads to several key recommendations for successful, routine analysis of water samples with special consideration for the determination of $\Delta^{17}O$.



## 2 Methodology

### 2.1 Analytical Protocol

A Picarro L2140-*i* cavity ringdown spectrometer was operated with the following configuration: an A0325 liquid autosampler
for injection into an A0211 vaporizer coupled to an A0214 micro-combustion module (MCM) which itself was coupled to the
L2140-*i*. The L2140-*i* and the A0211 both utilized A2000 diaphragm vacuum pumps (Vacuubrand #MD1). The autosampler
was equipped with a 10 µL syringe (Trajan #002982) that was manually cleaned between sequences using N-Methyl-2-
pyrrolidone (NMP, Fisher #AC390680010) by lubricating the plunger from the top of the syringe barrel with NMP, actuating
the plunger carefully until smooth movement was achieved, removing the plunger, submerging the plunger in NMP, wiping
the plunger with a cellulose wipe, reinstalling the plunger and repeatedly aspirating the NMP, and repeating the same process
using deionized water. No solvent rinsing was performed between injections, however each sample injection cycle dispensed
two 1.8 µL aliquots to waste prior to a 1.8 µL injection. The vaporizer used a 9.5 mm general purpose 'blue' septum (Trajan
#0418240) that was replaced after each sequence. The MCM requires a dry air carrier to perform combustion, which was
achieved using a cylinder of zero air (Airgas #AI Z300). The MCM contains a catalytic cartridge (Picarro #C0345) that ensures
complete combustion of organics and must be regularly replaced. When operating the instrument with the MCM off, the MCM
was always set to 'Warm' in order to prevent condensation of sample vapor within the MCM flow path. Fused insert vials
(Thermofisher #03FISV) were used for all injections, which were filled to 200 µL of their ~300 µL nominal (~400 µL actual)
volume and sealed with silicone/PTFE septum caps (LeapPalParts #009-13-8353) following Schauer et al. (2016).

Primary reference waters (VSMOW2 and SLAP2) were used for scale normalization from January 2019 to June 2020 to
establish acceptable performance of the instrument and to calibrate in-house laboratory reference waters and international
reference waters previously unconstrained for $^{17}$O composition (Table 1). In-house laboratory reference waters were selected
in order to 1) bracket the common range of $\delta^{18}$O and $\delta^2$H in natural waters across the globe and, in particular, to bracket low-
latitude precipitation and surface water samples which are routinely analyzed in our lab; and 2) capture a large range of $\Delta^{17}$O
values. Tap water from St. Louis, MO (STL), tap water from Big Sky, Montana (BSM), and bottled Kona drinking water from
Hawaii (Kona) were stored in 30L kegs following Tanweer et al. (2009). In addition, three 10 L polyethylene containers of
melted Antarctic ice core (ANT) were stored in a cold room for occasional analysis of more $^{18}$O- and $^2$H-depleted samples. All
secondary reference waters were measured independently for $^{17}$O composition at the University of Michigan via H$_2$O
fluorination and analysis by dual inlet-isotope ratio mass spectrometry (Table S1) using the same conditions as described by
Li et al. (2015). $\Delta^{17}$O values produced by our Picarro (Table 1) for control standards have a precision (1 standard deviation)
of 9-16 per meg (mean of 12). Our calibrated values are within error of the independent measurements (Table S1).



**Table 1. Reference materials used in this study. Values in bold are based on measurements by this**

| International References | Vials Analyzed[a] | $\delta^{18}O$ ( ‰ )[c] | $\delta^{17}O$ ( ‰ )[c] | $\delta^2H$ ( ‰ ) | *d*-excess ( ‰ ) | $\Delta^{17}O$ ( $10^6$ x $\delta$ ) |
|---|---|---|---|---|---|---|
| VSMOW2 | - | 0 | 0 | 0 | 0 | 0 |
| SLAP2 | - | -55.5 | -29.6986[e] | -427.5 | 16.5 | 0 |
| USGS45 | 210 | -2.238 | -1.1703[e] | -10.3 | 7.6 | 12 |
| USGS47 | 15 | -19.80 | -10.4642[e] | -150.2 | 7.9 | 40 |
| USGS50[b] | 11 | **4.8918[d]** | **2.5739** | 32.8 | -6.8 | **-6** |
| USGS53[b] | 37 | **5.4759[d]** | **2.8525** | 40.2 | -3.6 | **-35** |
| **Laboratory References[f]** | | | | | | |
| STL[b] | 182 | **-9.4161** | **-4.9627** | **-73** | **2.3** | **17** |
| Kona[b] | 459 | **0.1139** | **0.0527** | **1.2** | **0.3** | **-7** |
| BSM[b] | 51 | **-19.4850** | **-10.3022** | **-150.6** | **5.3** | **34** |
| ANT[b] | 1 | **-42.4484** | **-22.6379** | **-339.4** | **0.2** | **4** |

[a] Number of discrete vials analyzed by this lab. Excludes any vials used for scale normalization.

[b] Either in-house standards are reference waters whose 17O composition was previously unconstrained.

[c] Up to 4 decimal places are reported to allow reproduction of $\Delta^{17}O$. For traditional interpretation of delta values, we use and recommend rounding to two decimal places.

[d] We report our calibrated value rather than the recommended value for compatibility with our observed $\Delta^{17}O$. Additional, external analyses would be required to detect if the recommended value is subject to revision.

[e] Values derived from Berman et al., 2013.

[f] STL = St. Louis, MO, USA tap water; Kona = Kona Deep bottled drinking water; BSM = Big Sky, MT, USA tap water; ANT = ice core sample from Antarctica.

Sequences were structured to account for drift by bracketing the batch of unknowns with normalization and control standards, with an additional drift standard injected between every ~12 samples (Table 2). A warm-up vial was used consisting of our

drift standard, which, for the final 6 months of the 18-month measurement period, was spiked with a small amount of ethanol (0.022% v/v) and methanol (0.004% v/v) to serve as a quality check on the combustion performance of the MCM. Each normalization/control standard set (vial positions 3-6 and 51-54 of Table 2) was ordered from more positive to more negative $\delta^{18}O$ and $\delta^2H$ values. Aside from the warm-up vial (9 injections), all vials used 6 injections of the extended 'Long Pulse' injection routine resulting in a ~14.4 minute injection-to-injection duration and approximately 1.5 hours analytical time per

vial. Our typical sequence (Table 2) lasted ~3.3 days and was designed to allow some operator flexibility to ensure a regular schedule of 2 complete sequences (80 unknown vials) per week.



**Table 2. Typical Sequence Structure**

| Sample Type | Typical Standard | Vial Position | Number of injections |
|---|---|---|---|
| Warm-up / MCM QAQC | Alcohol-Spiked Kona | 1 | 9 |
| Drift Standard | Kona | 2 | 6 |
| Normalization Standard 1 | USGS53 | 3 | 6 |
| Control Standard 1 | USGS45 | 4 | 6 |
| Control Standard 2 | STL | 5 | 6 |
| Normalization Standard 2 | BSM | 6 | 6 |
| Drift Standard | Kona | 7 | 6 |
| Sample | | 8-20 | 6 |
| Drift Standard | Kona | 21 | 6 |
| Sample | | 22-35 | 6 |
| Drift Standard | Kona | 36 | 6 |
| Sample | | 37-49 | 6 |
| Drift Standard | Kona | 50 | 6 |
| Normalization Standard 1 | USGS53 | 51 | 6 |
| Control Standard 1 | USGS45 | 52 | 6 |
| Control Standard 2 | STL | 53 | 6 |
| Normalization Standard 2 | BSM | 54 | 6 |
| MCM QAQC | Alcohol-Spiked Kona | 1 | 6 |

Unknown samples run during the study period were predominantly unfiltered rainwater, although, ground-, tap-, and filtered river-water samples were run intermittently. Samples whose $\Delta^{17}O$ was intended to be measured were typically analyzed with a minimum of 3 replicates as discrete, 200 µL aliquots spread across separate sequences. See Sect. 4.3 for rationale.

## 2.2 Corrections for memory, instrument drift, and scale normalization

Sample-to-sample memory is a typical operating constraint of continuous flow instruments. The typically suggested approach to reducing memory for Picarro water isotope analyzers is to perform at least six injections and discard all but the last three (Picarro Inc., 2015b). A second approach is to perform an empirical correction by estimating the size of the memory reservoir(s) and using this information to remove the influence of the previous vial (Van Geldern and Barth, 2012; Gröning, 2011). We implemented both approaches in our standard processing routine: correction of all injections following the 'simple one-memory approach' of Gröning (2011) while also using only the last 3 injections of each vial for calculation of isotope values. Memory coefficients, defined as the fraction of the current vial's contribution to the observed isotope value, were determined using a 'memory terms sequence' using 5 sets of 25-injection replicates alternating between an enriched sample (Kona, $\delta^{18}O \approx 0$ ‰; Table 1) and a depleted sample (Antarctic ice, $\delta^{18}O \approx -42$ ‰). The last 8 injections of each were averaged to calculate the 'memory-free' values and then used with simple isotope mass balance to calculate the fraction (i.e., memory coefficient) of the previous vial at each injection (Gröning, 2011). Memory coefficients were relatively stable over time. Memory coefficients were updated by running the memory terms sequence every ~3 months.





Instrument drift was accounted for by the repeated injection of discrete vials of Kona (Table 1) spread throughout the sequence (Table 2). Given the length of our sequences (~3.3 days), the maximum daily drift according to specifications of 0.2 ‰ (for oxygen) and 0.8 ‰ (for hydrogen) can well exceed measurement precision of 0.025 ‰ and 0.1 ‰, respectively (Picarro Inc., 2017). Our drift correction was based on a linear regression of the drift standard delta values versus injection position, the latter of which is a proxy for time. To apply the drift correction, the slope of the drift regression was multiplied by an injection's position and subtracted from the injection's observed delta value. This was calculated and applied independently for $\delta^{18}O$, $\delta^{17}O$, and $\delta^{2}H$. Our L2140-*i* did not always exhibit linear drift and so the choice to perform the drift correction was made on a per-sequence basis.

Normalization to the VSMOW-SLAP scale was achieved by linear regression of the normalization standards. As the Picarro factory calibration is relatively stable over time, 'raw' $\delta^{18}O$ values are typically within ~1 ‰ of the corrected values, however these small variations result in the uncorrected $\Delta^{17}O$ being hundreds of per meg away from calibrated values (Figure S12). Uncorrected $\delta^{2}H$, however, has tended to drift directionally over time by ~5 ‰ (Fig. S12). Typically, USGS53 and BSM (Table 1) were used as normalization standards. Both the starting and ending sets of standards were used for scale normalization, yielding some averaging of instrument noise that may otherwise impact 'true' two-point linear normalization (Paul et al., 2007).

## 2.3 Processing

Post-run processing to apply the various corrections (Sect. 2.2) was performed using an R script (Supplemental File 1) following the approach of Schauer et al. (2016). The L2140-*i* exports 'coordinator data' of each injection as a comma-separated values (CSV) file whereas 'high-resolution data' is stored as Hierarchical Data Format (HDF) files containing instrument results at a frequency of ~1 Hz. In addition to a much higher resolution, the HDF files contain many more data streams useful for troubleshooting instrument behavior and performance. See Schauer et al. (2016) for greater detail about the L2140-*i*'s data types. After the user collected the appropriate private data for a sequence based on date and time, our processing script read the private data, used a set of criteria to find each injection (or 'pulse') based on $H_2O$ levels, and assigned them based on user input and our sequence template (Supplemental File 1). Each injection contained ~430 measurements based on the time to generate one line of high-resolution data (~1 Hz) and the duration of usable data (~8 minutes) during each injection cycle of the long pulse mode. Concentration ratios (R) of heavy-to-light isotopologues corresponding to each isotope system (Table S2) were used to calculated delta values expressed in per mil notation:

$$\delta = (R_{sample} / R_{standard} - 1) \times 1000 \tag{1}$$

where $R_{sample}$ corresponds to the observed signal and $R_{standard}$ corresponds to the observed R of an injection of VSMOW2 performed shortly after the instrument installation. The $\delta$ values correspond roughly, but not exactly, to the default,





uncalibrated δ values shown by the Picarro Data Viewer and Coordinator software during analysis. Schauer et al. (2016) noted that $\delta^2$H experienced increased memory experienced during long pulse mode and recommended using only the first ~200 seconds of $\delta^2$H data when integrating (via the arithmetic mean) each pulse while the oxygen delta values should use the full pulse. Our initial testing found that 180 seconds was optimal for $\delta^2$H, and our script used this value, although this may be an instrument-specific variable. Injections were then memory-corrected, summarized to the vial-level using the last 3 injections of each vial, assessed for drift correction, and then scale normalized. The derived values of deuterium excess (*d*) and $\Delta^{17}$O were calculated from the fully corrected isotope values. Deuterium excess was defined as:

$$d\text{-excess} = \delta^2\text{H} - 8(\delta^{18}\text{O}) \tag{2}$$

while $\Delta^{17}$O was defined as:

$$\Delta^{17}\text{O} = (\ln(\delta^{17}\text{O}/1000+1) - 0.528 \times \ln(\delta^{18}\text{O}/1000+1)) \times 10^6 \tag{3}$$

using a slope of 0.528 (Luz and Barkan, 2010) and expressed in per meg ($10^6 \times \delta$).

Our processing script produces an Excel file containing sheets that report various layers of data reduction: calibrated results of samples, summary statistics and results of quality control standards, injection-level and vial-level results corrected only for memory, and some additional diagnostic results and metadata. Sample results are output both rounded (for general reporting) and unrounded for easy recalculation of $\Delta^{17}$O values. Summary statistics for control standards include: observed arithmetic mean, observed standard deviation, the root mean square error, and mean signed difference. Root mean square error (RMSE) was calculated as:

$$\text{RMSE} = \sqrt{\frac{\sum_{i=1}^{n}(x_i - \hat{x}_i)^2}{n-1}} \tag{4}$$

where $x_i$ is the final, calibrated value of a standard and $\hat{x}_i$ is the current accepted value for that standard. The mean signed difference was calculated as:

$$\text{MSD} = \frac{\sum_{i=1}^{n}(x_i - \hat{x}_i)}{n} \tag{5}$$

following the same notation as RMSE. RMSE is used as the primary measure of precision and accuracy while MSD provides an estimate of bias from the accepted value. For comparison, we also provide an R script (Supplemental File 2) modified to operate on the Picarrro's coordinator output.





## 2.4 Statistical Methods

All data analysis and plotting was performed within R (R Core Team, 2017) using the "tidyverse" package set (Wickham, 2017). Picarro's HDF files were read using the "rhdf5" package (Fischer et al., 2020). All statistical hypothesis testing was performed by application of a balanced bootstrap approach using an appropriate sampling statistic (e.g., arithmetic mean, 210    ordinary least squares slope) and the statistic's 95% confidence interval was tested for overlap with zero to test for significance, which is equivalent to a *p*-value cutoff of 0.05 for null hypothesis testing (Davison et al., 1986). Unless otherwise noted, errors on summary statistics reported in Sect. 3 (given in brackets) are the 95% confidence interval.

## 3 Results

### 3.1 Memory Corrections

Memory coefficients generated by memory terms sequences as described in Sect. 2.2 show little variation over the measurement period (Fig. 1). $\delta^{18}O$ and $\delta^{17}O$ have nearly identical memory effects and averaged 98.8 [98.7 - 98.9] % of current vial contribution to the current pulse by the 4th consecutive injection from a vial, which falls slightly short of the stated efficiency of 99% by Picarro. This slight deviation may be due to the increased flow path introduced by the MCM device or due to our longer pulse duration than High Precision mode. As expected, $\delta^{2}H$ experiences greater memory and achieved 98.4 220    [98.3-98.4] % current vial contribution by the 4th consecutive injection, which slightly exceeds Picarro's stated performance of 98%. The shortened $\delta^{2}H$ data usage note in Sect. 2.3 may explain the improved performance. Our currently limited dataset (n = 2 sequences) for High Precision mode indicates performance for $\delta^{2}H$ matches the 98% specification (4th injection mean

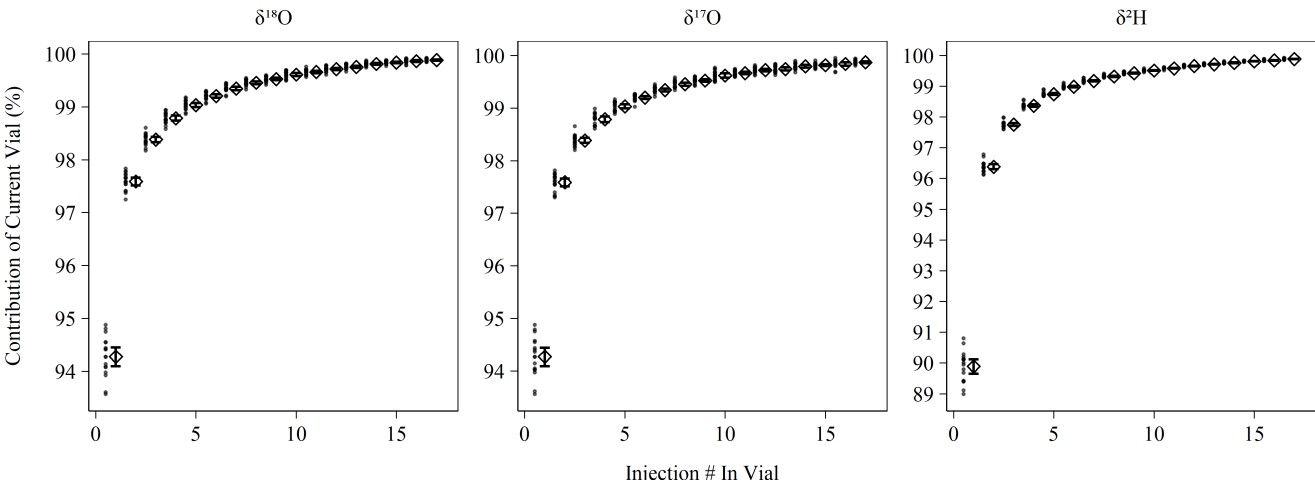

**Figure 1.** Percent contribution of the current vial's injection to the observed isotopic measurement based on memory terms sequences (section 2.2). Long-term mean and 95% bootstrapped confidence interval shown as diamonds and error bars, respectively, with contribution estimates from individual memory coefficient runs shown as filled circles. Note that $\delta^{18}O$ and $\delta^{17}O$ have different y-axis ranges than $\delta^{2}H$.





**Table 3. Long-Term Performance of Standard Reference Materials. Values for standard deviation (SD), root mean square error (RMSE) and mean signed difference (MSD) for all reference materials analyzed during the ~2 year measurement period.**

| International References | Vials Analyzed[a] | $\delta^{18}O$ (‰) | | | $\delta^{17}O$ (‰) | | | $\delta^2H$ (‰) | | | d-excess (‰) | | | $\Delta^{17}O$ ($10^6$ x $\delta$) | | |
|---|---|---|---|---|---|---|---|---|---|---|---|---|---|---|---|---|
| | | SD | RMSE | MSD | SD | RMSE | MSD | SD | RMSE | MSD | SD | RMSE | MSD | SD | RMSE | MSD |
| USGS45 | 210 | 0.0500 | 0.0503 | -0.0057 | 0.0279 | 0.0280 | -0.0031 | 0.3 | 0.3 | 0 | 0.3 | 0.3 | 0.1 | 12 | 12 | 0 |
| USGS47 | 15 | 0.0596 | 0.0639 | 0.0223 | 0.0319 | 0.0358 | 0.0156 | 0.6 | 0.6 | -0.1 | 0.5 | 0.5 | -0.2 | 12 | 12 | -2 |
| USGS50 | 11 | 0.0291 | 0.0301 | 0.0076 | 0.0169 | 0.0186 | -0.0074 | 0.3 | 0.5 | -0.4 | 0.3 | 0.3 | 0 | 8 | 10 | -5 |
| USGS53 | 37 | 0.0574 | 0.0579 | -0.0074 | 0.0323 | 0.0325 | -0.0035 | 0.3 | 0.7 | -0.6 | 0.3 | 0.6 | -0.5 | 12 | 12 | 0 |
| **Laboratory References** | | | | | | | | | | | | | | | | |
| STL | 182 | 0.0493 | 0.0540 | -0.0219 | 0.0273 | 0.0312 | -0.0150 | 0.3 | 0.3 | 0.1 | 0.4 | 0.5 | 0.3 | 11 | 11 | 0 |
| Kona[b] | 459 | 0.0639 | 0.0639 | 0.0003 | 0.0342 | 0.0342 | -0.0003 | 0.5 | 0.5 | 0.1 | 0.4 | 0.4 | 0.1 | 12 | 12 | -1 |
| BSM | 51 | 0.0510 | 0.0510 | 0.0000 | 0.0261 | 0.0261 | 0.0000 | 0.5 | 0.5 | 0 | 0.4 | 0.4 | 0 | 9 | 9 | 0 |
| **All Standards** | 965 | - | 0.0581 | -0.0046 | - | 0.0317 | -0.0033 | - | 0.5 | 0.1 | - | 0.4 | 0.1 | - | 12 | -1 |

[a] Number of discrete vials analyzed by this lab. Excludes any vials used for scale normalization.

[b] Kona vials were used for drift correction and should not be strictly interpreted as control standards.

of 98.0%). Correction for memory improves the RMSE of control standards in the case of all metrics except Δ17O where no difference was observed (Fig. S1). The magnitude of improvement was ~0.05 ‰ for δ18O and δ17O and ~0.5 ‰ for δ2H.

## 3.2 Drift Corrections

Instrument drift on the L2140-i is rated at a maximum 0.2 ‰/day for oxygen measurements and 0.8 ‰/day for hydrogen measurements. Our assessment of drift using the Kona standard as outlined in Sect. 2.2 indicated significantly less drift than the maximum specification during the 2-year measurement period (Fig. 2). Hydrogen measurements were almost always drift-corrected whereas oxygen measurements showed more variability and more often had sequences whose drift slope approached zero. However, for both oxygen and hydrogen measurements during the measurement period, the average drift slope was significantly different from zero as shown in the confidence intervals of Fig. 2. When combined with the sequence length of ~3.3 days, the magnitude of the daily drift rates (Fig. 2) exceeds typical instrument error (Table 3) by ~0.06 ‰ and ~0.4 ‰ for oxygen and hydrogen measurements, respectively.

In addition to our standard drift correction procedure, we tested several alternative approaches to drift correction. This was necessary in order to account for the multiple possible causes of instrument drift, which are not well understood but which would have different consequences for sequence structure and other practicalities of routine sample analysis. Further, the long-term drift slopes in Fig. 2 are all significantly different from zero, which indicates that the L2140-i – at least our specific unit – exhibits positive directional drift (i.e., more positive isotope values over time). On the time scale of individual sequences, drift is observed to vary beyond the long-term mean with oxygen measurements sometimes even exhibiting negative drift slopes (Fig. 2). The variability of short-term drift – i.e., the observed drift at the sequence level – may be due to extrinsic, time-





varying factors (e.g., environmental conditions) or it may be due to the fact that the magnitude of drift is comparable to short-term instrumental precision. If the former is true, then a series of drift standards with each sequence is necessary to capture the impact of these time-varying factors, as in our standard operating procedure. If the latter is true, then simply a large sample
size of sequences is necessary to estimate the 'true' drift terms (e.g., Fig. 2) and then these terms can be applied to each sequence without accounting for sequence-level drift standards. Finally, alternative approaches were tested to account drift that is non-linear and/or inconsistent over time.

To test these alternatives, we used the long-term coefficients from Fig. 2 to correct all the sequences during our measurement
period and calculated RMSE and MSD and found worsened performance compared to our standard drift-correction procedure for most isotope measurements (Table 4). We also reprocessed sequences by drift-correcting using linear interpolation between individual drift standards, which would better account for non-linear drift. Linear interpolation between individual drift standards also produced worsened performance for most isotope measurements (Table 4). As an alternative to our standard procedure of applying a drift correction only when the drift standards vary directionally, we also reprocessed sequences
completely leaving out the drift correction, by always applying the drift correction, and by always applying the drift correction calculated from only the first and last bracketing drift standards (Table 4). Always applying sequence-level linear drift correction using either all the drift standards or only the bracketing first and last drift standards provides results very close to our standard procedure (Table 4), although this should be expected as our standard procedure typically applies the sequence-level linear drift correction. Long-term performance of $\Delta 17O$ was insensitive to the method of drift correction.

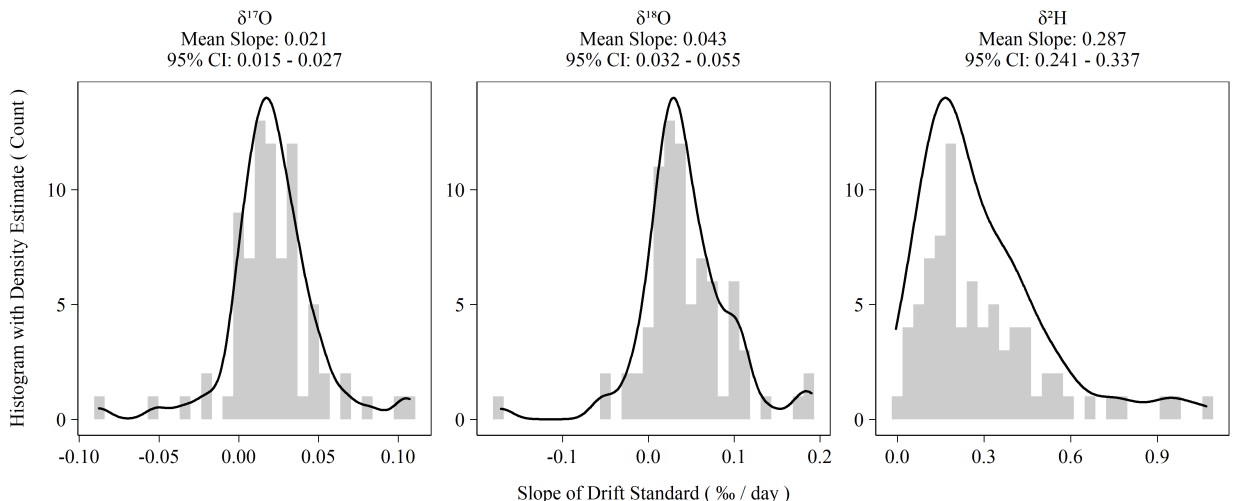

**Figure 2.** Histograms of ordinary-least-squares regression slopes of drift standard isotope values versus elapsed sequence time. Summary statistics are presented above each plot. The 95% confidence intervals are the 2.5 and 97.5 percentiles of the bootstrapped distribution of mean slopes.





**Table 4.** Accuracy metrics for all control and drift vials of known isotope composition for various drift correction approaches. Values for root mean square error (RMSE) and mean signed difference (MSD; in parentheses) for all vials analyzed during the ~2 year measurement period.

| Drift Correction | $\delta^{18}O$ ( ‰ ) | $\delta^{17}O$ ( ‰ ) | $\delta^2H$ ( ‰ ) | d-excess ( ‰ ) | $\Delta^{17}O$ ( $10^6$ x $\delta$ ) |
|---|---|---|---|---|---|
| Standard Procedure (see Section 2.2) | 0.058 (-0.005) | 0.032 (-0.003) | 0.5 (0.1) | 0.4 (0.1) | 12 (-1) |
| Application of Drift Coefficients from Figure 2 (see Section 3.2) | 0.093 (-0.005) | 0.050 (-0.004) | 0.5 (0.1) | 0.7 (0.1) | 13 (-1) |
| Linear Interpolation Between Drift Standards (see Section 3.2) | 0.089 (-0.009) | 0.047 (-0.005) | 1.0 (0.2) | 0.6 (0.2) | 12 (-1) |
| No Drift Correction | 0.107 (-0.007) | 0.057 (-0.005) | 0.5 (0.1) | 0.7 (0.1) | 13 (-1) |
| Always Apply Sequence-Level Linear Drift Correction | 0.067 (-0.006) | 0.036 (-0.004) | 0.5 (0.1) | 0.5 (0.1) | 12 (-1) |
| Always Apply Bracketed Drift Correction (see Section 3.2) | 0.068 (-0.004) | 0.037 (-0.004) | 0.5 (0.1) | 0.4 (0.1) | 12 (-1) |

## 3.3 Contamination by Organic Compounds

Organic contamination during the last 6 months of the measurement period was monitored as described in Sect. 2.1 with the use of a sample of our Kona standard spiked with amounts of ethanol and methanol following Picarro's recommendations for assessing MCM cartridge health for $\delta^{18}O$ and $\delta^2H$ in the user manual (Picarro Inc., 2015a). The MCM manual suggests using a 'simulated plant water' solution ranging from ~1.3 % (v:v) alcohols to ~0.26 % (v:v) alcohols, equivalent to 10,648 mg / L and 2,130 mg / L, respectively. In these cases, $\Delta^{17}O$ is elevated by well over 1000 per meg. The concentration employed in our MCM quality assurance standard as described in Sect. 2.1 is equivalent to a 50-fold dilution (~213 mg / L) of the original 1.3 % solution and results in a $\Delta^{17}O$ elevation of ~100 per meg without use of the MCM (or when the MCM cartridge has failed). Elevation of $\Delta^{17}O$ due to spiked alcohols is detectable from the unspiked standard in as little as 42 mg / L, or ~250-fold diluted from the original 1.3 % alcohols solution. The threshold for detectable alteration of the measurement is similar for other isotope measurements (Fig. 3). An MCM cartridge was considered spent when the MCM quality assurance standard exceeded +100 per meg relative to the pure Kona standard, which was always observed to occur as a single step rather than a partial failure over a series of injections. However, as the quality assurance standard only bracketed sequences, we do not

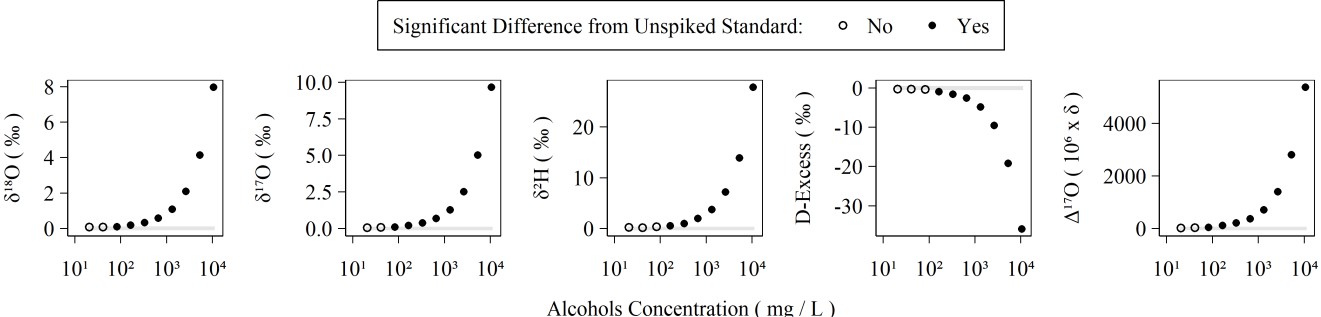

**Figure 3.** Observed isotopic measurements with increasing amounts of alcohols spiked into our Kona lab standard. The highest concentration here corresponds to 1.1 % (v:v) ethanol and 0.2 % (v:v) methanol and was serially diluted in 50% steps to generate the remaining points. Each concentration was measured using two replicate vials with 6 injections each. The average and standard deviation of the last 3 injections of each vial was compared to triplicate vials of Kona standard using a Monte Carlo approach that simulated normal distributions from the observed averages and standard deviations and then subjected to a balanced bootstrap unpaired test of differences in the mean values and evaluated at the 95% confidence interval level.





know the exact failure mode of the cartridges except that it occurs over the duration of a single sequence. During the 6 months

of routinely operating with the MCM on, we exchanged 5 cartridges. Each cartridge lasted between 5 and 75 days with an

275 average lifetime of 31 days. Therefore, the extreme sensitivity of $\Delta^{17}O$ to organics contamination makes the effective cartridge

lifetime much shorter than the expected four months of operation when analyzing only $\delta^{18}O$ and $\delta^2H$ (Picarro Inc., 2015a).

We tested and developed several techniques for flagging samples with suspected organics contamination. First, in some cases,

visual inspection of the data is sufficient to flag samples of possible concern because the measurements lie well outside the

280 observed natural range of $\Delta^{17}O$ in meteoric waters (approximately –50 to + 60 per meg, although values are most commonly

positive; Aron et al., 2021). Second, the L2140-$i$ produces diagnostic values based on spectral characteristics to help the user

determine if organic contamination has occurred. However, Picarro's ChemCorrect software is unable to correct for

interference and moreover does not operate on $^{17}O$-mode data (Picarro Inc., 2015b). Our standard procedure used these

suggested values to flag samples that may be contaminated. Third, we developed a potentially more sensitive metric for

flagging potentially contaminated samples that makes use of the 18O-Laser (referred to hereafter as "18O Laser Flag"): We

compared the standard deviation of the instrument's two $\delta^{18}O$ values corresponding to spectral peak ratios (see Table S2) of

11 / 2 (used for 17O-mode $\delta^{18}O$ measurement) and 1 / 2 (used for normal-mode $\delta^{18}O$ measurement but still operated during

17O-mode). During sequence processing, we calculate the mean and standard deviation of 18O-Laser values for standards and

apply a 2-standard-deviation threshold for flagging samples as potentially contaminated.


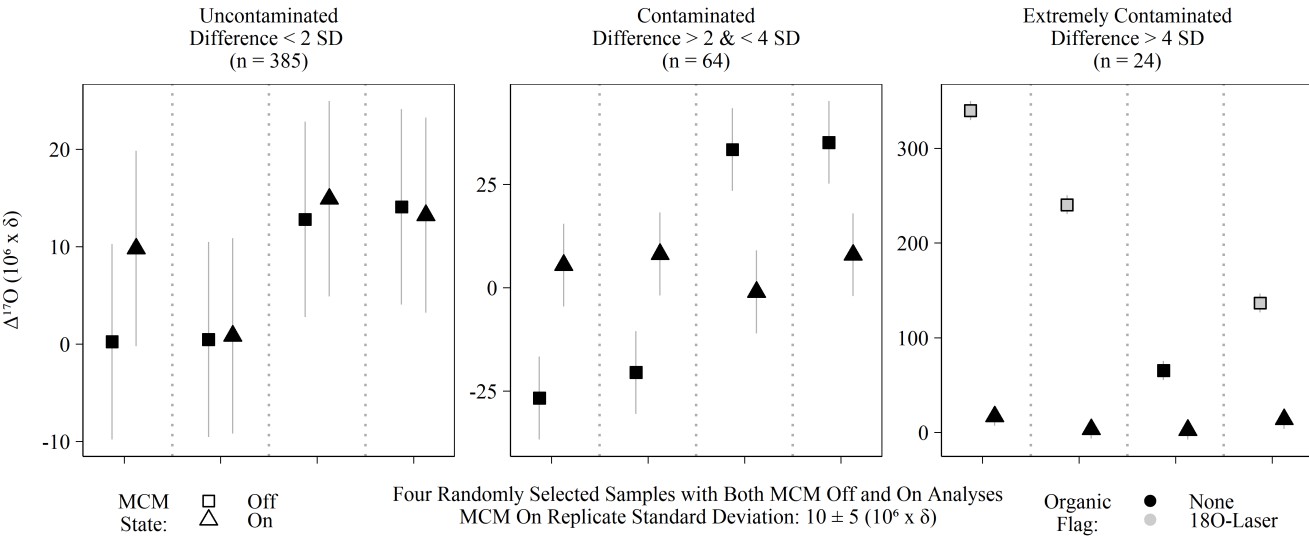

**Figure 4.** Randomly selected subset of calibrated Δ17O of samples run with the MCM off and on categorized into three groups based on the differences between the MCM off and on analyses. Differences exceeding 2 standard deviations of instrumental precision are assumed to derive from organic contamination. Only 'Extremely Contaminated' samples are typically identified by spectral flags as having organic interference (grey-filled points).





The MCM is a peripheral device recommended by Picarro when users intend to analyze samples with substantial organics interference, such as plant waters. Initially, we operated the instrument with the MCM off, for two reasons: First, we were measuring pure rainwater and tap water samples and therefore did not expect significant organic contamination; and second, we expected any minor contamination would be detected via Picarro's spectral contamination flags. This initial batch of 'MCM

off' samples (n = 473) contained 16 calibrated $\Delta^{17}O$ values that greatly exceeded the expected natural range (as high as 628 per meg). Those unusual samples were nearly always (15 out of 16) flagged by our 18O-Laser metric when the $\Delta^{17}O$ exceed 100 per meg, but was only flagged by Picarro's suggested metrics in extreme cases (> 500 per meg). Due to this variable flagging, we reran replicates from this batch with the MCM on and found that $\Delta^{17}O$ for obviously contaminated samples (i.e., both spectral flags and extreme $\Delta^{17}O$) was then shifted to within the expected natural range: the full range of 'MCM on' $\Delta^{17}O$

values was -61 to 58 per meg. False negatives – 'MCM off' samples with no contamination flags but excessively different $\Delta^{17}O$ values from their 'MCM on' replicates – were determined by comparing $\Delta^{17}O$ differences between MCM modes and used either a 2 or 4 standard deviation threshold of 20 or 40 per meg, respectively, based on instrumental precision. Differences between 20 and 40 per meg were categorized as 'contaminated' whereas differences > 40 per meg were categorized 'extremely contaminated'. Only 10 of the 24 'extremely contaminated' samples were spectrally flagged and none of the 64 samples from

the 'contaminated' group of Fig. 4 triggered any type of spectral flags. All 'extremely contaminated' samples had elevated $\Delta^{17}O$ whereas the 'contaminated' group was roughly split between positive and negative biases. While some of the 'contaminated' group (14% of MCM off samples) may simply be uncontaminated outliers, only 4% of individual MCM on replicates were greater than 20 per meg away from their replicate means, which is consistent with 2 standard deviations accounting for ~95% of a normal distribution. All of the 473 samples compared in this Sect. were rainwater samples that we

would have no *a priori* reason to expect organic contamination.

### 3.4 Long- and Short-term Precision and Accuracy

The long-term performance for standard reference materials on the L2140-*i* within the measurement period is summarized in Table 3. The long-term precision of standards (i.e., the standard deviation of final, calibrated values) was essentially identical to our primary measure of accuracy (RMSE) due to the relatively small bias in accuracy as measured by MSD. Long-term

summary statistics can mask some variability in sequence-to-sequence performance, so we also summarized standards at the sequence level (Fig. 5). Sequence-level mean values (Fig. 5, Table S3) for RMSE and MSD are essentially identical to long-term performance (Table 3). However, short-term bias in accuracy can be a much larger term than long-term bias, with the standard deviation of MSD at the sequence-level about 50% the size of RMSE (Fig. 5, Table S3). This effect is not typically problematic for primary isotope measurements due to the already small error, but for $\Delta^{17}O$ the standard deviation of sequence-

level MSD (Fig. 5, Table S3) is 6 per meg, which can indicate systematic error at the sequence level. The USGS45 standard was included as a control in all sequences as its $\Delta^{17}O$ has some consensus and its calibrated values all converged on the current accepted values (Table 3) with consistent performance through time (Fig. S4).





The duration of the measurement period of replicate samples had a small, but weak influence on replicate precision. For
measurement of $\Delta^{17}O$ in unknown samples, we utilized discrete vials measured across multiple sequences, an approach that
provides a measure of medium-term precision (i.e., reproducibility across a limited set of sequences) and produces optimal
mean errors (see Sect.s 4.3 and 4.5). The average standard deviations of unknown samples for each measurement type (Fig.
S3) are equal to or exceed the long-term performance of standards (Table 3). The use of discrete sequences for unknowns
resulted in a mean measurement period of 7 (range: 0 – 21) months as defined by the first and the last time an unknown was
measured. There were weak, but significant relationships of unknown replicate precision versus the measurement period for
all isotope metrics (Fig. S5) and all relationships became insignificant when the data were more equally weighted using 2-
week binned average standard deviations (Fig. S6). When using the regression coefficients from Fig. S5, the shortest
measurement period to excess long-term standards performance (Table 3) was 18 months for $\delta^{17}O$. Our sample storage strategy
of 4 mL vials using polyethylene 'PolyCone' caps wrapped with Parafilm stored in a refrigerator at 4 °C is apparently effective
for at least this storage duration.

The standard deviation of the last 3 injections of a vial was used as measure of short-term precision. We used this metric to
assess overall short-term precision across the entire measurement period (Fig. S7, n = 4112 discrete vials) and to evaluate the
impact of the instrument's two measurement modes and pulse length on short-term precision. The varying operation modes
(normal mode versus $^{17}O$ mode) and pulse length routines (high precision versus long pulse) were compared by running
replicate sequences (n = 48 vials each) in each of the modalities and assessed using paired differences. In $^{17}O$ mode, the results
of this experimental comparison (Fig. S8) indicated that the long pulse routine improves precision for $\delta^{17}O$, $\delta^{18}O$, and $\Delta^{17}O$

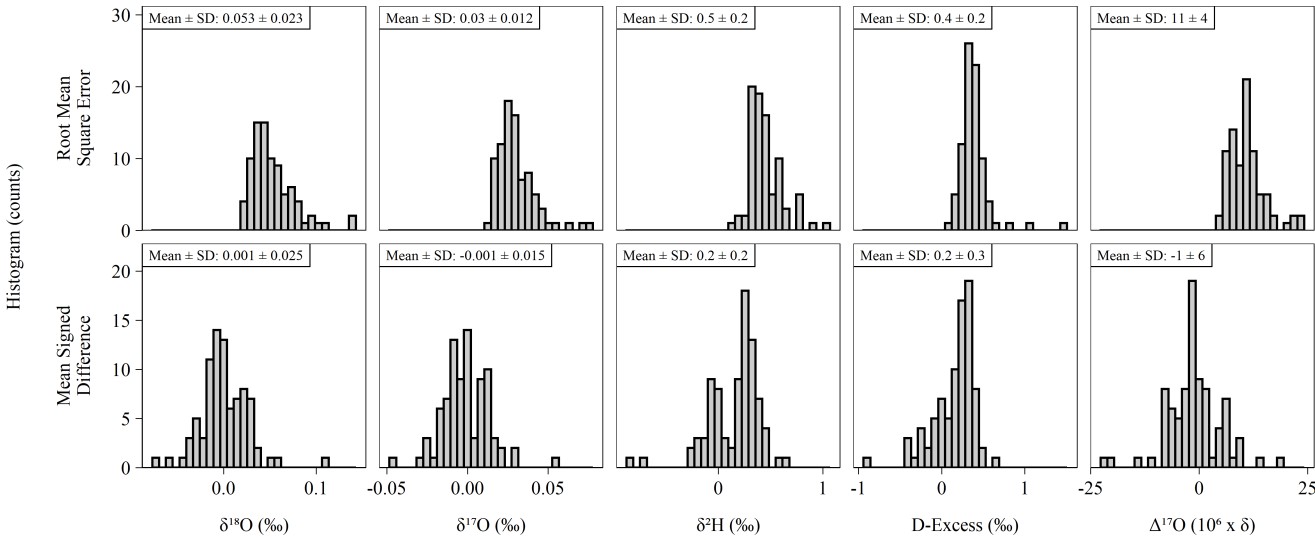

**Figure 5.** Histograms of accuracy metrics (see section 2.3 for formulas) for all sequences (n = 85) run during the study. Note that RMSE
cannot be less than zero. Means and standard deviations (SD) shown above each histogram for each pairing of isotope measurement and
accuracy metric.





relative to the high precision routine. Pulse length did not produce significant improvements in short-term precision of $\delta^{18}O$ when operated in normal (i.e., $^{17}O$-disabled) mode (Fig. S8). When comparing $^{17}O$-enabled vs. normal modes (Fig. S9),

operating the instrument in normal ($^{17}O$-disabled) improved the precision of $\delta^{18}O$ an average of by 0.010 [0.007–0.012] ‰ whereas $^{17}O$-enabled improved the precision of $\delta^2H$ by an average of 0.15 [0.12–0.17] ‰ with no differences between modes for *d*-excess. The short-term precision during this limited experiment was similar to that across the entire measurement period (Fig. S7).

Finally, we assessed the influence of data source (Picarro's high-resolution data vs. coordinator data) on precision. Our standard data processing approach used the Picarro's high-resolution output via our R processing script (Supplemental File 1). An alternative processing approach (Supplemental File 2) uses the Picarro's default user-accessible coordinator data output via a modified R script written to ingest the coordinator output's simple CSV files. In principle, the only differences between these methods were that the high-resolution approach used our own pulse-detection parameters (versus those hard-coded into

the Picarro to generate coordinator data) and that our high-resolution data script uses only the first ~180 s of each pulse for determining $\delta^2H$ versus the entire pulse for coordinator data (Schauer et al., 2016). All corrections (memory, drift, and scale normalization) were calculated the same. To compare these, we processed ~6 months of sequences from the first half of 2020 (16 standard sequences with 3 memory terms sequences) using both R scripts. The high-resolution processing approach improves short-term precision by 3.9, 2.4, and 6.5 % over coordinator data (Fig. S10) for $\delta^{17}O$, $\delta^{18}O$, and $\Delta^{17}O$, respectively.

These values were calculated using the mean absolute differences between the processing approaches and dividing by the precision of the high-resolution results. In contrast to short-term precision, standards with known isotopic composition had statistically indistinguishable RMSE values for all measurements except $\delta^2H$ (Fig. S11).

## 4 Discussion and Recommendations for Operational Procedures

### 4.1 Corrections

In our standard protocol, we apply three such corrections: memory, drift, and scale normalization. Of these, the only correction commonly understood to be necessary is that of scale normalization – required to place the uncalibrated data on the internationally accepted VSMOW-SLAP scale (Paul et al., 2007). Corrections for memory and drift are commonly applied by users of laser-based isotope instruments (Chesson et al., 2010; Van Geldern and Barth, 2012), although various approaches are possible (Berman et al., 2013) and some analytical conditions can be maintained to avoid needing these corrections (Schauer

et al., 2016). However, the necessity of corrections is determined by the level of precision and/or accuracy needed by the end-user and their research question. The addition of standards to measure and account for these corrections consumes both analyst and analyzer time and, thus, the choice to apply them must balance investment of time against requirements of instrument performance. While the application of post-analysis corrections to data are necessary, such corrections should be minimized to prevent 'over-correction,' i.e., introduction of bias and/or artefacts (e.g., over-fitting of low signal-to-noise relationships,



incorrectly modelling the function to be accounted for). Here, we discuss the impact of each of these corrections and their relative importance for different users and applications.

Memory correction was determined by an empirical isotope balance mixing model as described in Sect. 2.2 following Van Geldern and Barth (2012). Memory coefficients determined this way showed little variation over the 2-year measurement
period (Fig. 1) and significantly improved all isotope measurements except $\Delta^{17}O$ (Fig. S1). Other users have chosen different approaches to handling instrument memory. Efforts to avoid memory correction include increasing the number of discrete measurements in a vial or the ordering of measurements to ensure adjacent measurements are isotopically similar (Schauer et al., 2016), which in both cases minimizes the impact of sample-to-sample memory. The former effect can be seen in Fig. 1 where memory is increasingly diminished with consecutive measurements with the primary tradeoff being an increase in
analyzer time spent on a single vial. The latter effect of 'isotopic ordering' is only possible if approximate isotopic values are known *a priori*, which is only possible in certain situations (e.g., directional measurement of ice cores) or with preliminary isotopic measurement. In our laboratory, our measurements typically target meteoric waters that vary widely on an event-to-event basis and would require preliminary measurement to order. Even then, this approach would still be problematic because ensuring small isotopic differences between all adjacent unknowns may not be possible for any given batch of samples. We
therefore find the method of calculating and applying memory coefficients to be practical as well as effective in minimizing sample-to-sample memory for routine analysis of unknown meteoric waters with a wide range of variability. Calculation of memory coefficients are done through a memory terms sequence as described in Sect. 2.2 measured on a ~3-month interval while application of coefficients is done during post-processing. Aside from the requirement of determining the memory coefficients, this does not add significant analysis time as this correction does not require positions in a standard sequence to
be applied.

Drift correction used a sequence-level linear regression slope of drift standards as described in Sect. 2.2 and following standard practice in continuous flow applications. Correction for drift requires that the user include a series of replicates of a standard to measure the observed drift during a sequence. Our standard sequence structure uses 5 drift replicates (Kona; Table 2) and,
thus, consumes approximately ~9% of a sequence's run time. The benefits of drift correction are predominantly in improving the accuracy of $\delta^{18}O$, $\delta^{17}O$, and d-excess with RMSE approximately halved compared to not drift correcting (Table S3). However, there is no consistent effect observed for either $\delta^{2}H$ or $\Delta^{17}O$. The lack of consistent improvement for $\delta^{2}H$, which itself has the largest daily drift values (Fig. 2), is surprising and suggests that the larger errors inherent with $\delta^{2}H$ measurement may outweigh the effect of drift. Considering the range of isotopic variability in natural samples, the improvement to RMSE
is substantial for $\delta^{18}O$ (0.107 ‰ uncorrected vs. 0.054 ‰ corrected; Table S3) and perhaps less meaningful (or nonexistent) for the other quantities. These results suggest that omitting drift correction may be an appropriate decision if the end-user is accepting of higher error for $\delta^{18}O$, $\delta^{17}O$, and d-excess. The opportunity cost of drift correction is not negligible: If our typical sequence structure (Table 2) were adjusted to exchange the drift standards for unknown samples, then our overall throughput





of unknowns would be increased by 12.5%. An appropriate compromise might be including drift standards only in the
beginning and end of each sequence ("bracketed drift correction" in Table 4), which performs only marginally worse than our
standard procedure while also still increasing sample throughput compared to our typical sequence by 7.5%. The primary
drawback of this compromise is that the operator has little visualization of the drift effect during the sequence and would be
obligated to simply always apply the slope calculated from the two drift standards.

Scale normalization is mandatory to ensure compatibility of interlaboratory measurements. Additionally, measurements made
within a laboratory but separated in time benefit from increased comparability via scale normalization. However, we note that
there may be some unique circumstances or applications for which scale normalization would not be strictly necessary. For
example, the device could be used for measuring artificially enriched isotopic tracer samples whose differences are expected
to exceed long-term variation. In our lab, the long-term instrumental drift on the Picarro L2140-$i$ is surprisingly small for $\delta^{18}O$
and $\delta^{17}O$ (~2 ‰ and ~1.5 ‰ ranges, respectively) with directional drift of ~6 ‰ for $\delta^2H$ (Fig. S12). Due to the sensitivity of
$\Delta^{17}O$, its long-term variation is extreme with a range of ~600 per meg. Therefore, measurement of pulses of highly enriched
isotope samples could yield largely satisfactory results without scale normalization, although such results would necessarily
not be on the VSMOW-SLAP scale except in the loosest sense.

**4.2 Error structure of replicate analyses and implications for Δ17O**

The relative magnitude of the natural range of meteoric $\Delta^{17}O$ (~110 per meg, Aron et al., 2021) to the short-term precision of
the measurement (~11 per meg) – a ratio of 10 to 1 – is distinct from the other data streams produced by the L2140-$i$, which
have range-to-precision ratios at least an order of magnitude greater than $\Delta^{17}O$. This may be an exaggeration when considering
samples from a specific locality where the range of observed values is much smaller than the global range. However, for
example, $\delta^{18}O$ would need a natural observed range of only 0.164 ‰ to match the 10 to 1 ratio of $\Delta^{17}O$.


In other analytical settings, such as with isotope ratio mass spectrometry, a common approach to overcoming issues of precision
is to perform repeat measurements and report their final average (Berman et al., 2013). If each measurement is a sample from
a distribution with shape dictated by the performance of the instrument, then the average of repeated measurements will
approach the average of the distribution, which itself approximates the 'true' value if bias is sufficiently small (Miller and
Miller, 1988). In the case of the L2140-$i$ (and likely other CRDS instruments), the sequence-level bias (MSD) has variability
equal to approximately 50% the long-term RMSE (Fig. 5). As such, distributing replicate measurements within a single
sequence would not be adequate for approaching the 'true' value. Therefore, we choose to distribute our replicate
measurements across distinct (i.e., independently calibrated) sequences in an approach similar to many IRMS and some IRIS





applications (Uechi and Uemura, 2019). As the

average bias of sequences is close to zero (Table 3, Fig. 5), this approach should effectively minimize the impact of bias. While our approach takes more effort than preparing replicate vials for measurement in a single sequence, for our scientific

purposes the impact on error minimization is well worth the effort, and we can be much more confident that the mean of our replicates minimizes sequence-level accuracy bias.

Figure 6 demonstrates the effect of replicate measurements using all control and drift standards analyzed during the study period. The decrease in error (i.e., width of the confidence interval) with increasing number of averaged vials follows the

trend in the standard error of the mean. Using only the n = 1 data from Fig. 6, we can predict the observed error structure of increasing replicates according to the SEM (Fig. S13) with all $r^2 > 0.99$, which strongly indicates our error structure is

normal. Figure 6 can be used to estimate the number of replicates needed to achieve a certain error threshold for unknown samples. Other users seeking to reproduce our performance should observe a very similar error reduction gradient as

shown in Fig. 6, assuming comparable long-term instrumental performance. For $\Delta^{17}O$, we choose to measure three independent replicates in our

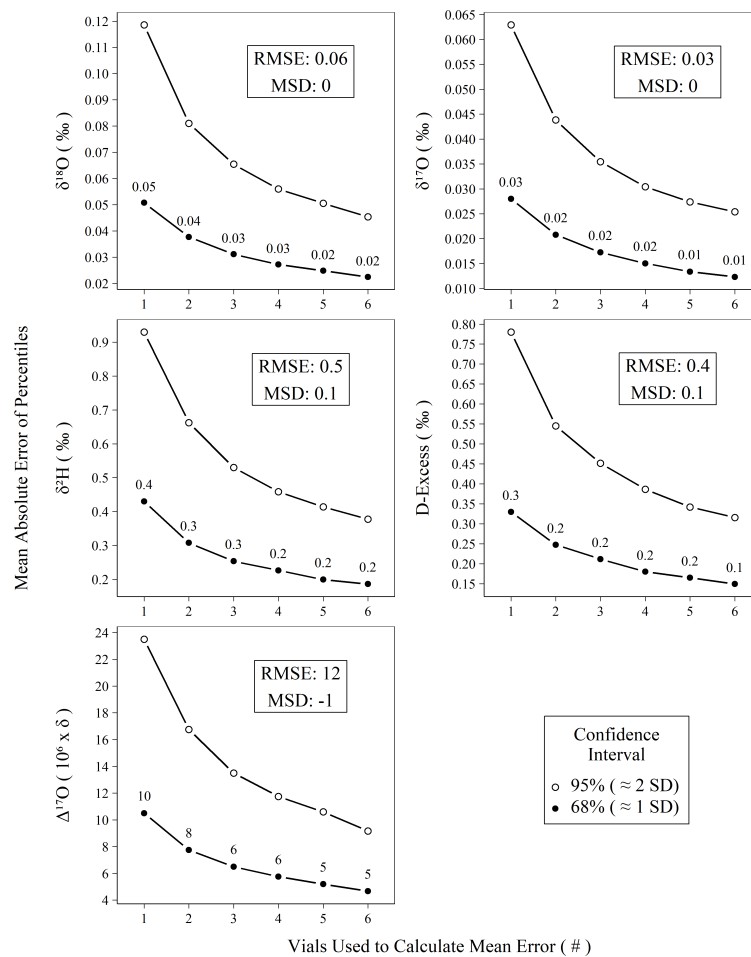

**Figure 6.** Mean absolute error of calibrated standards for isotope measurements versus the number of vials averaged prior to mean error calculation. All measurements of control and drift standards (n = 896) were differenced from their accepted values, resampled (n = 100,000) without replacement into replicates of varying (1 – 6) size, and summarized as means to create probability distributions of mean errors. Confidence intervals (68% and 95%) were calculated for the probability distributions and the mean absolute values of their percentiles are plotted against their replicate size. The two intervals chosen, 68% and 95%, roughly correspond to one and two standard deviations, respectively.

standard operating procedure, which yields an error of ~6 per meg from the 'true' value for the 68% confidence interval. Although this is particularly important for planning measurement structure for $\Delta^{17}O$ due to its limited natural range, the results

in Fig. 6 can also be used to if increased confidence is desired for other metrics.



### 4.3 Sensitivity to Organic Contamination

The sensitivity of IRIS instruments to certain dissolved, volatile organic compounds – typically short-chain-length alcohols – is well-known and is especially problematic for analysis of plant and soil waters (Brand et al., 2009; Martín-Gómez et al., 2015; Nehemy et al., 2019; West et al., 2010a). Picarro's MCM device was developed to remove interfering compounds via
combustion and Martín-Gómez et al. (2015) demonstrated effective removal of short-chain alcohols as long as their concentrations are below ~2% v:v, although others have found mixed results and opt for offline methods to minimize organic interference (Chang et al., 2016). In the presence of organic contamination, spectral interference causes $\delta^{18}O$, $\delta^{17}O$, and $\delta^2H$ to shift by a few to tens per mil with the magnitude of the shifts depending on the identity and concentration of the organic contaminants (Brand et al., 2009). Both Picarro's and our own 18O-Laser data quality flags readily detect organic
contamination in the form of ethanol/methanol mixtures with total concentrations exceeding 1% (v:v) of water. However, the water isotopologues' absorption spectra used by the L2140-*i* are quite narrow (Steig et al., 2014) compared to the wide absorption spectra of both ethanol and methanol (Adachi et al., 2002; Dong et al., 2019). The apparent effect is that each isotopologue, and, thus, isotope ratio measurement, is distinctly affected by these organic contaminants. This effect is magnified for $\Delta^{17}O$ as it is both based on two isotope ratios and is interpreted at the per meg level ($10^6$) rather than per mil
($10^3$). Thus, an alcohol contamination of 1.1 % ethanol and 0.2 % methanol shifts $\delta^{18}O$ and $\delta^{17}O$ by only ~3 ‰ but over 2000 per meg for $\Delta^{17}O$ (Fig. 3). In reality, these shifts are comparable when placed on similar scales (i.e., 2000 per meg is equal to 2 ‰), but the practical sensitivity is realized at the level that the measurement is scientifically interpreted at.

The MCM appears to effectively remove organic contamination, but with one important caveat: The catalytic elements in the
combustion cartridge are expended over time, and the effective lifespan of a cartridge is far shorter for $\Delta^{17}O$ analyses than for $\delta^{18}O$, $\delta^{17}O$, or $\delta^2H$ analyses on their own. The testing procedure recommended by Picarro only extends to ~2100 mg/L alcohols, however $\Delta^{17}O$ is sensitive to alcohol contamination to approximately ~40 mg/L alcohols (Fig. 3). Our approach of analyzing an alcohol-spiked water sample before each sequence is crucial because it enables us to be positive the MCM's catalysts are effective prior to the analysis of samples. If sequences are not being run continuously, we typically also run the alcohol-spiked
MCM quality control sample at the end of a sequence, too, to ensure the catalyst was functional throughout the run. Additional alcohol-spiked quality control samples could be run throughout a sequence, but we choose to avoid this to preserve the activity of the MCM catalysts.

The behaviors documented in Fig. 3 and Fig. 4 strongly suggest that effective organics removal is mandatory for reliable
measurement of $\Delta^{17}O$ in all types of meteoric water sample, even rainwater, using the L2140-*i*. Without confident removal of organics, samples can be shifted away from their true values while remaining well within the range of natural variability as well as being essentially undetectable through current spectral flagging techniques. The same is true of all other isotope metrics (Fig. 3), but, while the magnitude of those shifts do exceed analytical error at similar levels of dissolved organics, they are





much more rarely interpreted near the limits of analytical error. However, users that are interpreting other isotope metrics at
or near the limits of analytical error should employ organics removal to ensure that their unknowns actually have the same
analytical error as pure standards.

The levels of dissolved alcohols required to shift analytical measurements is between ~ 40 and ~ 80 mg / L (Fig. 3), which is
equivalent to ~20 to ~40 mg C / L. The amount of dissolved organic carbon (DOC) in rainfall globally tends to vary at levels
well below this at between 0.2 and 11.4 mg C / L (Iavorivska et al., 2016). Our low-latitude rainfall samples would need to
have at least double the upper range of global rainfall DOC for simple alcohols to be the source of our spectral interference.
This may be plausible, as a daily-resolution record of precipitation from a site in São Paulo, Brazil found an average
precipitation DOC 20% higher than the Iavorivska et al. (2016) global synthesis and rain-event-scale measurements as high as
50 mg C / L (Godoy-Silva et al., 2017). However, while simple alcohols are common constituents of leaf water, precipitation
can contain many different types of volatile organics, including terpenoids associated with volatile emissions from plants
(Guenther et al., 2006), aromatic hydrocarbons associated with biomass or fossil fuel combustion (Abdel-Shafy and Mansour,
2016), and additional, poorly characterized compounds (Altieri et al., 2009). If the L2140-*i* is more sensitive to other compound
classes found in precipitation than the simple alcohols tested here, then the threshold for spectral interference may be even
lower than observed in Fig. 3.

**4.4 Comparison with Δ17O Performance in DI-IRMS and other IRIS approaches**

IRIS devices have, historically, been considered less technically demanding and time consuming than IRMS, either continuous
flow or dual inlet (Berman et al., 2013; Wassenaar et al., 2018). However, our results here agree with other reports (Van
Geldern and Barth, 2012; Gröning, 2011; Pierchala et al., 2019; Wassenaar et al., 2021) that operator choices and attention to
corrections can greatly attenuate the performance of IRIS devices in much the same ways as IRMS. The primary differences
are analytical throughput, cost (both purchasing and maintenance), and the technical skill required for operation and routine
maintenance. At least for Picarro IRIS devices, the training needed for operation and routine maintenance is rather simple, as
most hardware failures that occur within the device require that the unit be repaired and recalibrated by Picarro technicians.
However, post-analysis corrections, such as those detailed in this paper, are as necessary for IRIS devices as they are for IRMS
to yield reproducible results on a common, international reference scale.


In terms of accuracy and precision, the long-term accuracy of our L2140-*i* (12 per meg overall; Table 3) is comparable to or
better than both IRIS and DI-IRMS performances reported in the recent literature (~8 per meg via DI-IRMS (Berman et al.,
2013); 8 - 21 per meg via IRIS (Schauer et al., 2016, Pierchala et al., 2019). While our long-term $\Delta^{17}O$ performance is slightly
worse than Schauer et al. (2016), our analysis of precipitation and tap waters required that we overcome memory between
isotopically disparate adjacent samples, which is a common issue with IRIS devices (Van Geldern and Barth, 2012; Gröning,



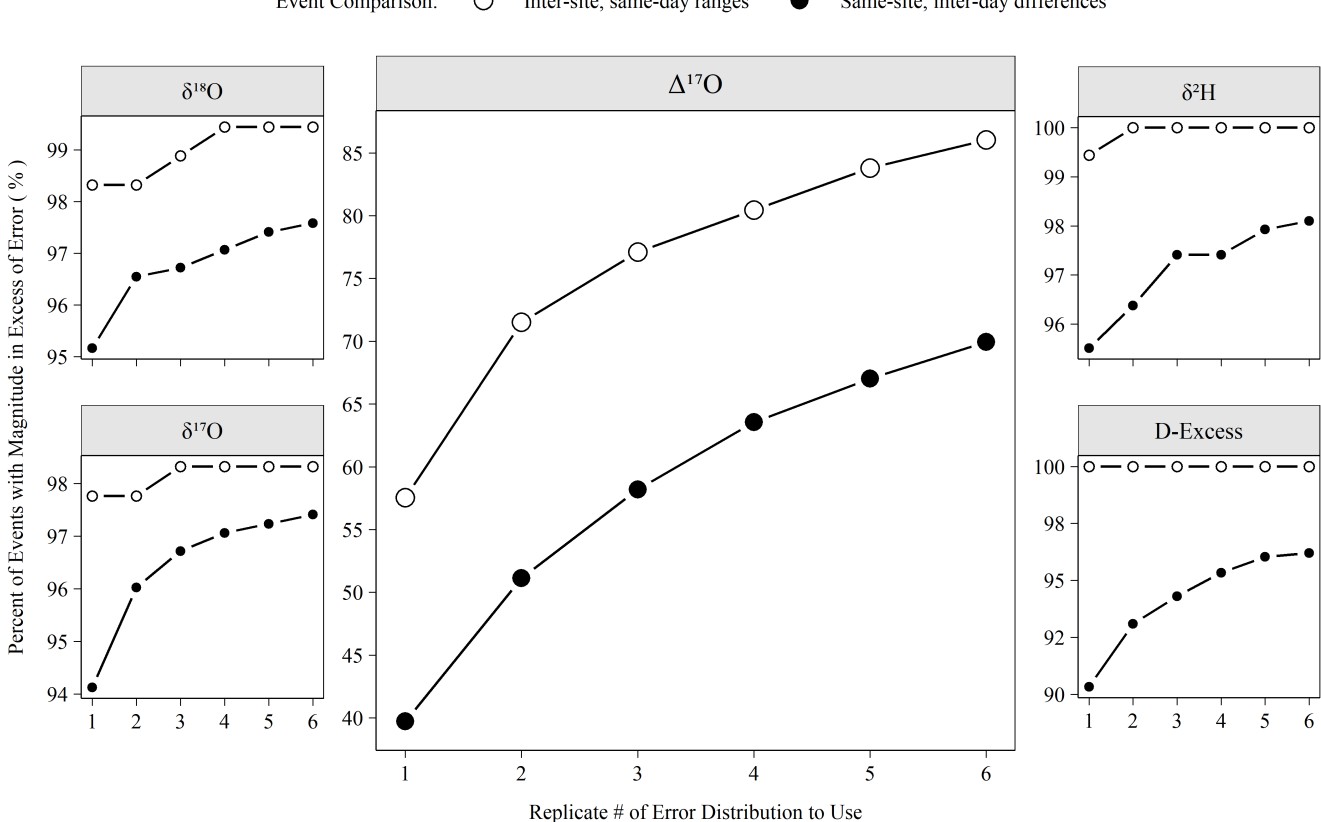

**Figure 7.** The percentage of meteoric precipitation 'events' captured by varying numbers of analytical replicates. Inter-site, same-day ranges is the range of all sites from a Ugandan monitoring network (see section 2.1). Same-site, inter-day differences are the differences of measured isotope values from sequential precipitation events (< 7 day span). If the event magnitude (either range or difference) exceeded the error for a given replicate number (Figure 6), then it is considered here as 'detected'.

2011; Lis et al., 2008). The throughput of our approach (~80 unknown vials / week) is comparable to best practices in triple oxygen isotope IRIS and DI-IRMS techniques (Barkan and Luz, 2005; Berman et al., 2013; Pierchala et al., 2019; Schauer et al., 2016).

Beyond short-term precision and assessment of accuracy bias, practically any of these instrument types can be utilized to achieve comparable results. In the case of the L2140-*i*, we show that accuracy metrics for meteoric water samples can rival those of DI-IRMS as long as an appropriate number of replicates is chosen, as we show empirically in Fig. 6. Our choice of 3 distinct replicates requires a total of ~6.3 h per unknown when sequence structure is considered. This throughput could be improved (or worsened) depending on operator demands for final error metrics by adjusting the number of discrete replicates

that are utilized for each unknown according to Fig. 6. In the case of $\Delta^{17}O$, our method approaches the ~8 per meg measurement precision of common DI-IRMS approaches (Berman et al., 2013) when two replicates are used, (~4.2 h) albeit with a slightly worse throughput (Barkan and Luz, 2005).





The number of replicates to perform is a critical operational choice as this sets the analytical throughput of the instrument. Fig.
7 demonstrates the effect of replicate number on a potential scientific question. Utilizing the low-latitude precipitation samples
that were the bulk of unknowns run during the period considered here, we defined two types of 'events' that we might want to
detect. Inter-site, same-day ranges is the range of isotope values observed at all sites with rain on a given day in the network
and same-site, inter-day differences is the difference of isotope values observed between consecutive (< 7 days apart)
precipitation events at a single site. If the magnitude of the range or difference exceeded the replicate error structure shown in
Fig. 6, then the 'event' was 'detected'. Figure 7 shows that the large majority of these 'events' are readily detectable (i.e.,
exceed our error estimate) for any replicate number for all measurements except $\Delta^{17}O$. For $\Delta^{17}O$, an increase from 1 to 3
replicates allows for the detection of ~15 % more events in absolute terms or 20 % (ranges) to 30 % (differences) in relative
terms. Although the scientific importance of such events is another question to be investigated, these events must first be
detected to study their importance.

**4.5 Operational Choices to Optimize Performance of $\Delta^{17}O$, $\delta^{18}O$, $\delta^{2}H$, and *d*-excess in Meteoric Water Samples using the Picarro L2140-*i***

Our results demonstrate that various operation modes and user choices for the Picarro L2140-*i* require tradeoffs between data
quality, time, effort, and/or training. Some operational choices that we describe above can optimize performance of one
measurement while degrading the quality or precluding measurement of another. For example, our limited experimentation
with 17O-modes suggests that 17O-disabled mode results in better performance for $\delta^{18}O$ but surprisingly slightly worse
performance of $\delta^{2}H$, and of course precludes analysis of $\Delta^{17}O$. Other user choices can clearly benefit performance of all isotope
variables with only minimal additional time and effort, such as using high-resolution for post-processing (see below), whereas
other choices disproportionately improve some variables more than others, such as the approach to drift correction, so the
decision to invest the extra time should depend on the scientific questions being investigated. That said, there are several
operational choices that we contend are necessary for reliable and reproducible analysis of $\Delta^{17}O$, such as removal of organics.
In this section, we offer recommendations for the particular use-case of (1) analyzing predominantly natural, meteoric waters
where large sample-to-sample differences are expected, and (2) desiring optimal performance of $\Delta^{17}O$ without sacrificing the
quality of $\delta^{18}O$, $\delta^{2}H$, or *d*-excess. We expect this is a common use case for many laboratories wishing to incorporate $\Delta^{17}O$
analyses into existing hydrologic, atmospheric, biological, and geological investigations based on stable isotopes in water, or
for new investigators wishing to analyze $\Delta^{17}O$ in novel settings.

**Corrections for memory, drift, and scale normalization.** We contend that all of our corrections (memory, drift, and scale
normalization) should be performed in order to optimize data quality. However, as discussed in Sect. 4.1, the number of drift
standards could be reduced to a "bracketed" approach as long as some sacrifice to the performance of $\delta^{18}O$ is acceptable to the
user.



**Long Pulse vs. High Precision mode.** We recommend the increased analysis time of the Long Pulse mode because it significantly improves precision for $\delta^{18}O$, $\delta^{17}O$, and $\Delta^{17}O$ relative to High Precision mode, consistent with Schauer et al. (2016). The shorter pulse lengths of High Precision mode would require more replicates to match the performance of Long

Pulse mode. While we do not have sufficient data from High Precision mode to evaluate its error versus replicate pattern, if the pattern is similar to Long Pulse mode (Fig. 6) in that error is reduced as a function of the inverse root of replicate number, then it would require ~5 replicate analyses to achieve the same performance as our standard procedure of 3 replicates in Long Pulse mode. This would take approximately the same amount of analysis time but require substantially more user preparation, use more standard and sample material, and actuate the syringe much more often. As noted by others (Van Geldern and Barth,

2012; Schauer et al., 2016), syringe failure is by far the most common reason for sequence failure and reducing the number of actuations is typically desired.

**Processing data using High-Resolution vs. Coordinator streams.** Although a post-processing choice and not an operation mode, we observed significant, but essentially negligible improvements in precision for $\delta^{18}O$, $\delta^{17}O$, and $\Delta^{17}O$ when processed

using high-resolution versus coordinator outputs (see Sect. 2.3 for details on output types). However, high-resolution processing strongly outperforms coordinator data in terms of $\delta^2H$ accuracy (~0.25 ‰ RMSE improvement, Fig. S11), which is due to the shorter $\delta^2H$ integration time that is made possible by working on the 1 Hz-scale, high-resolution output that further reduces the impact of sample-to-sample memory (Schauer et al., 2016; Steig et al., 2014). Working with the high-resolution h5 files requires the use of some sort of command line-based program capable of ingesting the h5 format (e.g., R, Matlab,

Python) as well as navigating the date-time organized high-resolution folder structure. Combined, these make working with the high-resolution output more onerous than the much simpler, sequence-level summary CSV files of the coordinator output that can also be processed using available graphical user interface approaches (e.g., Coplen and Wassenaar, 2015; Gröning, 2011). The high-resolution output is also rich in additional diagnostic data streams such as the ability to calculate the more sensitive 18O-Laser spectral contamination metric (Fig. S2). Our unit has also exhibited occasional errant scans where only a

single line (~1 Hz) of high-resolution data has a poor spectral fit and extremely divergent isotope readings bounded by otherwise normal readings (Fig. S14). While we believe this particular problem may be unique to our unit, other unknown problems with these devices may be present and may only be detectable through analysis of the data-rich high-resolution files. We do not recommend using coordinator output unless the worsened $\delta^2H$ accuracy is acceptable.

**Number and sequencing of replicate measurements.** Replicate number should be determined based on the needed estimated accuracy of the measurement. Figure 6 is an effective guide assuming similar long-term performance as our device. If your performance is better (or worse), then you should consider doing a similar analysis on your own data for more accurate estimates of error. We choose to distribute our unknown replicates across distinct sequences and recommend this approach to other users based on reasoning discussed in Sect. 4.2.




**Removal of organic matter via the MCM**. Our results using rainfall indicate that online removal of organic contaminants is mandatory to ensure data quality. Nearly 20% of our samples (Fig. 4) exhibit symptoms of organic interference with much less (3%) being detectable by spectral contamination flags, which is an experience confirmed by other users (Chang et al., 2016). Off-line removal using activated charcoal or solid-phase extraction can remove some organic contaminants, but

typically remove only about 90 % of the starting concentration (Chang et al., 2016). This may be suitable for samples already near the limits of spectral interference (Fig. 3), although our limits are only for short-chain alcohols common in leaf extracts and may not represent organics found in rainfall. Future work on the concentration and specific identity of these contaminants will be useful in guiding strategies to handle organic inference for IRIS analysis.

## 5 Conclusions

In this work we present a measurement scheme and ~2 years of analyses using a Picarro L2140-*i* to measure all the stable isotopes of natural (predominantly meteoric) waters with a focus on optimized measurement of $\Delta^{17}O$. While isotope scale normalization is obviously mandatory, we find that our recommended post-processing corrections for instrumental drift and sample-to-sample memory only strongly improve $\delta^2H$, $\delta^{17}O$, $\delta^{18}O$, and *d*-excess, whereas relatively little benefit is found for $\Delta^{17}O$. Critically, $\Delta^{17}O$ is shown to be extremely sensitive to organic spectral interference and that this interference is often not

detected by spectral contamination flags. The MCM is marketed by Picarro as an optional device, but the sensitivity of $\Delta^{17}O$ to organics indicates that organics removal is required for confident measurement of any natural waters that may contain volatile organic carbon, including rainwater collected in field settings. However, the catalyst lifetime of MCM cartridges is quite variable and there is no automatic indication of its failure. We resolve this by including a quality control standard intentionally spiked with interfering short-chain alcohols to ensure effective organics removal by the MCM.


We note that the uncertainty of $\Delta^{17}O$ occupies a much larger fraction of its natural variability than other water isotope measurements. While our approach performs comparably with other laser-based devices (Pierchala et al., 2019; Schauer et al., 2016), we find that the variability of calibration bias for a sequence (Fig. 5) is a critical factor in producing accurate measurements of unknown samples. This is overcome by distributing replicates of unknown samples across distinctly

calibrated sequences and we measure this effect on accuracy empirically using control standards. For our recommended approach of 3 replicates, a total of ~6.3 h per unknown sample is required accounting for standards and inter-sequence downtime and yields mean absolute errors of 0.3 ‰, 0.03 ‰, 0.02 ‰, 0.2 ‰, and 6 per meg for $\delta^2H$, $\delta^{18}O$, $\delta^{17}O$, *d*-excess, and $\Delta^{17}O$, respectively (Fig. 6). Due to replication, these are less than long-term RMSE (Table 3, Fig. 6). Our measurement approach and post-processing steps are applied in conjunction with modifications such as increased pulse-length and shorter

integration times of $\delta^2H$ as described by Schauer et al. (2016).



Most of our recommendations are relatively easy to implement. For $\Delta^{17}O$, we find that most post-processing is unnecessary and that the only critical features for accurate and precise measurement is sufficient integration time (either by increased injections or longer pulses) and distribution of analytical replicates cross distinctly calibrated sequences. Our finding of $\Delta^{17}O$
organics sensitivity is specific to the L2140-*i* but, given similar spectra, likely impacts any infrared laser device. For overall performance of the instrument, we do find drift and memory corrections are necessary. While these post-processing steps can be onerous, memory correction removes the need to isotopically 'order' samples. We document two avenues for data export from the instrument with appropriately matched processing scripts written in R. The use of the default coordinator output is, undoubtedly, more user-friendly than the h5-based high-resolution stream, with the primary analytical benefits of high-
resolution data being slightly improved precision of oxygen-isotope measurements (Fig. S10) and improved accuracy of $\delta^2H$ (Fig. S11). We provide standard operating procedures for post-processing complete with example data for both output types (Supplemental Files 1 & 2).

The recent WICO2020 intercomparison exercise (Wassenaar et al., 2021) clearly demonstrated the apparent difficulty of
making accurate and precise $\Delta^{17}O$ measurements by laser spectrometry. This difficulty was present despite the lack of any organic-spiked samples (employed in WICO2016; Wassenaar et al., 2018) that would have caused much more serious deviations in $\Delta^{17}O$. Excepting organic interference, we demonstrate that the primary weakness for laser spectrometry $\Delta^{17}O$ is sequence-level calibration bias. Our presented strategy overcomes this and yields comparable performance and throughput to DI-IRMS. This is achieved through a suite of operational parameters, sequence structure, and post-processing corrections, but
provide some options to ease adoption. Although the increased adoption of triple-oxygen measuring laser spectrometry devices has expanded greatly in recent years, operator skill and care is required to produce robust $\Delta^{17}O$ measurements that are competitive with DI-IRMS. The accessibility of laser spectrometry combined with careful operation will help rapidly expand the study of the complete stable isotopic composition of water and enable the detection of signals previously hidden in the noise.

**6 Data availability**

Data used in the presented analyses and figures is contained in Supplemental File 3. At the time of publication, the samples used as unknowns form the basis of an ongoing research project and have been anonymized.

**7 Supplement link**

The supplement can be found by DOI (10.17605/OSF.IO/HGN8K) or URL (osf.io/hgn8k) and contains the supplemental
figures and tables. Supplemental files 1 and 2 contain instructions, file structures, and R scripts for post-processing of Picarro



results. Supplemental file 1 is used for the post-processing of high-resolution data and supplemental file 2 is used for the post-processing of coordinator data. Supplemental file 3 contains all the data used in the presented analyses and figures.

## 8 Author contribution

JAH and BLK equally designed the experiments and JAH carried out the bulk of laboratory work. JAH performed statistical analyses and prepared the manuscript with close collaboration with BLK.

## 9 Competing interests

The authors declare that they have no conflict of interest.

## 10 Acknowledgements

We gratefully acknowledge those who provided or assisted in the collection of the samples that were run during the 18-month period of this study. This includes: the Institute of Tropical Forest Conservation, the Uganda Wildlife Authority, ChildVoice International, and members of the Ruhija, Buhoma, Karuma, Paraa, and Gulu communities in western Uganda; Jeremy Diem, Shamilah Namusisi, Karen Bailey, Jonathan Salerno, Joel Hartter, Mike Palace, Pamela Grothe, Kathleen Elliott, Zach Eilon, Samantha Stevenson; funding for sample collection or analysis from University of Mary Washington, the Washington University International Center for Energy, Environment, and Sustainability, National Science Foundation GSS #1740201, National Geographic Society #HJ-097R-17, and a David and Lucile Packard Foundation Fellowship in Science and Engineering to B. Konecky. We thank Twila Moon and Peter Neff for providing waters for use as laboratory standards.

## 11 Financial Support

Financial support for this work was provided by NSF-GSS #1740201, the Washington University Earth and Planetary Sciences department, and a David and Lucile Packard Foundation Fellowship in Science and Engineering to B. Konecky.

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
