# Peer review of "Optimization of a Picarro L2140-i Cavity Ring Down Spectrometer for Routine Measurement of Triple Oxygen Isotope Ratios in Meteoric Waters"

_Atmospheric Measurement Techniques, 2022_

## Author Response (AR1)

Dear Dr. Janssen,

Please see the following pages for our final author reply. Our reply consists of the detailed responses we already posted with the addition of the line numbers of those changes as a secondary bullet. Please note that the line numbers of these changes correspond to the tracked changes version of the resubmitted manuscript so that our changes are clearly visible. Please let me know if any additional information or content is needed. Thanks!

Best,

Jack Hutchings

Reviewer #1:

- All author responses are bulleted. For completeness, the original reviewer text is included here in full.

**The manuscript "Optimization of the Picarro L2140-i Cavity Ring Down Spectrometer for Routine Measurement of Triple Oxygen Isotope Ratios in Meteoric Waters" by JA Hutchings and BL Konecky takes a close look at the way many of us use this instrument and the way we analyze the data. The paper is extremely thorough and takes the time to test many aspects that all users of this model, and really all IRIS and even IRMS instruments, should make note of. The manuscript is well written, well organized, sufficiently referenced and warrants publication in AMT after considering a few minor suggestions described below.**

- Thanks again for the thoughtful and thorough review of our manuscript.

**The permeg notation: "10^6 * delta" is awkward and confusing. I am not familiar with expressing permeg in this way. The authors define a delta value using equation 1 which includes the *1000 term. To me, then, this means delta values are in permil and have thus had 1000 multiplied by them. So if Delta17O is 10^6 * delta, then it is * 10^9. I know this is not what the authors mean, but perhaps you can see my point. I suggest simply using "per meg" and removing the "10^6 * delta" throughout. Equation 3 is sufficient for the authors definition of D17O.**

- This is a fair point. We'll go ahead and use 'per meg' going forward.
    - 195-200, but through the manuscript, figures, and tables.

**18O Laser Flag: The 18O Laser Flag seems like a great tool. I would like to see the authors make a short statement about how they came upon the "18O Laser Flag". The authors also need to describe in greater detail how the flag value is calculated, or is it simply the "SD_18_18" value from the h5 files? What specific values of the 18O Laser Flag constituted action? That is, what is the threshold value of the 18O Laser flag that, if exceeded, meant organic contamination? Why does this 18O Laser Flag work? Are the authors assuming the alcohol absorption will have a changing baseline from 7190 to 7200 /cm ? It seems to that the 18O Laser Flag is a way of comparing the the two lasers. 18O Laser Flag and D17O both appear to be good indicators of contamination and both make use of comparing the two disparate absorption bands. I suppose we could also use deuterium excess calculations that use 18O from laser 1 and 18O from laser 2. I wonder then if it is appropriate to extend a generalized statement about the utility and benefit of having two lasers to compare. I realize much of this is musing speculation but it would be great to have a sentence or two giving the reader insight into the authors thought process around this idea and this flag. As it is, the 18O Laser Flag is just thrown in without any introduction or elaboration. Lastly, and very minor, on line 285, the authors state "referred to hereafter as "18O Laser Flag"" but rarely if ever refer to this metric in that way. I see "18O-Laser Values", "18O Laser", "18O-Laser Metric", "18O-Laser spectral contamination metric".**

- As mentioned in our open discussion response, our revised manuscript will expand on the '18O Laser Flag' metric. We also see our sloppy reference in the text/figures to the metric, which will be rectified in the revised manuscript. Beyond what was already said in our open discussion response, we should clarify that the reason we left the concept somewhat hanging is that, although it appears to be a more powerful indicator of spectral contamination than the default

metrics, the '18O Laser Flag' is ultimately found to be insufficient to detect minor amounts of contamination that must be dealt with by removal of the interference. The interference must, then, be removed before the sample is analyzed. We accomplish this through online combustion using Picarro's micro combustion module, but offline removal is possible, although efficiency of techniques such as solid phase extraction appear to be about 90% removal (Chang et al., 2016), which may not be sufficient for samples with high enough initial contamination. We will add detail on the inferred operating principle of this metric as well as its limitations in the revised manuscript.

- o Additional text at 290-300 as well as 490-500.

**What is the outcome or course of action after finding an aberrant point such as is shown in Figure S14? Does use of h5 level data allow you to recover without rerunning the water? Given the authors preference for high resolution data over coordinator data, further explaining this process will help their case. Minor comments related to this topic: Fig S14 (a) caption, I think should refer to injections, not vials, not because the statement isn't true as is, but because the authors are showing injection data, not vial data. Also, and again, minor, it may be worth pointing out in the caption of S14 that C and D data do not come from the h5 files referred to thus far in the manuscript, but rather from the very high resolution spectral data. I know the authors already say "spectral readings" but helping the readers who don't use any of the h5 files may be useful.**

- We addressed this in our open discussion response and will add additional context in the manuscript. We'll also update our reasoning for using high resolution processing in light of this feature as well as the '18O Laser Flag'.
  - o Additional text at 620-525.

**Technical Comments:**

**- Consider referring to the supplemental figures and tables as appendices. It seems as though AMT uses supplemental information to refer to a text based document and stand-alone figures and tables are appendices.**

- We had done this originally and were informed that appendices are, in AMT, appended to the end of the paper such that all the supplemental figures/tables would actually be part of the manuscript. Since we have a good number of additional figures/tables, we opted to treat them as supplemental so that the formatted manuscript would not be excessively long/large due to all the extra figures/tables.

**- Check for delta notation missing superscripts throughout the manuscript, for example line 216.**

- Yes, there were a few! We think we have them all now.

**- Also, on occasion, I see "D-Excess" instead of "d-excess". Change to lowercase "d" throughout.**

- Thanks – we'll double check this throughout the manuscript.

**- Line 39 - "slope of 0.528 calculates values" is awkward. Reword.**

- Fixed, thanks!
  - o Removed extraneous text at line 40.

**- Line 59 - "isotopogues" should be "isotopologue"**

- Fixed, thanks!
    - Now line 58.

**- Line 66 - "H2H16O" should be "1H2H16O" with appropriate superscripts, of course.**

- Indeed, thanks!

**- Table 1 caption ends prematurely. "...measurements by this study." perhaps?**

- Yes, you are correct. That was cut off when we imported the table.

**- Line 154 - "To apply the drift correction, the slope of the drift regression was multiplied by an injection's position and subtracted from the injection's observed delta value." As worded, this sentence suggests that the slope is multiplied and then subtracted. I think the slope is multiplied by the injections position and then the product is subtracted?**

- Correct! Fixed that text.
    - Changed text at line 161.

**- Line 182 - "experienced increased memory experienced", I think, should read "experienced increased memory"**

- Fixed, thanks!
    - Removed the extraneous word at line 188.

**- Line 246 - "account drift" should read "account for drift"**

- Fixed, thanks!
    - Added 'for' at line 252.

**- Line 317 - "term", while correct, is slightly awkward with the two other "-term" uses in that sentence. Perhaps "factor"?**

- Yes, thanks!
    - Line 327.

**- Line 333 - The word "excess" seems out of place. "exceed" perhaps?**

- Correct, thanks!
    - Line 343.

**- Line 337 - "as measure" should perhaps read "as a measure"**

- Correct, thanks!
    - Line 347.

**- Figure 5 - The average of the MSD should be zero, or not different from zero. I am not sure if it is worth showing this in the figure text. It would be like changing your equation 5 to x_i minus x_bar instead of x_hat. We expect the sum of (x_i minus x_bar) / n to equal zero or not different from zero. Also, and minor, I am not sure you need an additional y-axes label of "Histogram (counts)" since it is in**

**the caption. Furthermore, Table S3 has exactly the same data presented as the text in Figure 5. I feel the same way about the mean MSD in Table S3 and since you have the same data in Fig 5, I suggest deleting Table S3.**

- We're not completely sure we follow your reasoning. Are you saying that, ideally, our MSD values *should* be zero / indistinguishable from zero? And that we should highlight this feature of the metric? If so, then we have added some text to that effect. We also agree that table S3 is an exact reproduction of the text in Figure 5, but we reasoned that some readers would prefer tabular data and that there isn't any harm in having it in the supplement.
  - Added some clarifying text in the Figure 5 caption located between lines 348 and 349.

**- Line 345 - "an average of by" should perhaps read "by an average of"**

- Correct, thanks!
  - Line 355.

**- Figure 6 caption - "were differenced" should read "were subtracted"**

- Fair enough, we've changed that for clarity.
  - Figure 6 caption near line 470.

**- Line 500 - "sample" should be "samples"**

- Correct, thanks!
  - Line 513.

**- Line 628 - "only strongly" seem to contradict each other. Seems like "only" should be deleted**

- Agreed.
  - Line 645.

**- Figure S1 - The order of figures is inconsistent with other figures. All other figures are d18O, d17O, dD, d-excess, D17O.**

- Fair. This was an issue in a couple other figures (S10, S11) and they are all now in the same order as other figures.

**- The data in Figure S3 seems like it may be better presented as histograms, which are made good use of elsewhere in the manuscript.**

- We wanted to preserve the original individual data points in this figure to demonstrate that lack of extreme outliers that might otherwise be masked by a histogram.

**- Figure S13 caption - "differenced" should be "subtracted"**

- Fair enough, we've changed that for clarity.

**- Table S2 - missing peak 1, H218O. Also, consider removing this table entirely and simply cite Steig et al 2014.**

- Good catch! For completeness we will leave the table in, especially since it is not in the main text and does not occupy more valuable space.

Reviewer #2:

- All author responses are bulleted. For completeness, the original reviewer text is included here in full.

**The manuscript presents an in-depth investigation of potential problems and an improvement in detecting and correcting these through optimizing the analysis of the raw data and adjusting the measurement series setup. This problem is particularly relevant for the secondary isotope parameter D17O, as this parameter is strongly influenced by spurious spectra of various organic molecules, as the authors show using water standards spiked with alcohol. I like their approach and was surprised that they even observed significant shifts of D17O in rainwater due to such influences. The manuscript should be published in AMT as it discusses important problems with this analytical method and suggests remedies that should be known in the scientific community.**

- Thanks again for the thoughtful and thorough review of our manuscript.

**Major points:**

**It would be interesting to know how the samples were stored. Which containers (material, colour), caps and membranes were used and how long were they stored before analysis? This could be important for the development of organic substances in the sample container.**

- We note this in the manuscript (around line 335 in section 3.4): 4 mL glass vials with polyethylene 'PolyCone' caps wrapped with Parafilm and stored in a 4°C refrigerator. We'll clarify that our vials are made of glass in the revised version. This is, to our knowledge, a common approach and should not appreciably leach any organics, although we suppose it may be somewhat possible given that the liner is made of plastic.
  - Additional text located at 135-138.

**Since the analysis time for high-precision measurements per sample - as suggested by the authors - is very long at 6.3 hours, it would make sense to first run the sample value range with a 1-injection sequence and order them according to the pre-measured values. This would minimize the memory effects and could therefore lead to a shorter total time.**

- We addressed this a bit during open discussion. To expand: We do not believe we would see any benefit for our use-case by doing this. The samples are typically analyzed in triplicate only because we seek to minimize the estimated error around the D17O measurement, which itself is largely unaffected by memory correction (Fig. S1).
  - Note that some of the discussion at lines 388-405 is relevant to this comment.

**Why was the BSM and not the ANT standard used as the second normalization standard, the latter standard would be most negative for both the d18O, d17O and the dD that you have available. Your choice requires extrapolation of your normalization for highly depleted samples, but I suspect that you have not measured any such samples other than the ANT standard.**

- In fact, we have begun using the ANT standard as our depleted normalization standard as we have moved into measuring more temperate-climate waters with our device. However, for the mostly low-latitude samples discussed in this paper, the less-depleted BSM standard was more appropriate.

**It is not clear how the memory effect for D17O was calculated, as this cannot be calculated directly from a simple D17O balance, as it relates to a difference in logarithmic values. Please add a sentence on how this was done.**

- This is addressed in the open discussion and we'll clarify the point in the main text in section 2.2.
  - Clarification at lines 151-153.

**The memory effect has been calculated based on alternating the Kona and ANT standards, i.e. an enriched and a depleted isotope value from which percentage memory influences are calculated. This is correct, but this leads to different times for reaching the instrument noise level as the differences among the samples will be significantly smaller for all isotopes. Therefore, an ordering according to their pre-measured or estimated sample values could most probably lead to a significant reduction of the overall time investment without losing precision. See also a minor comment to Fig. 6 below.**

- We addressed this partially in our open discussion response. We agree that, if injection amounts could be changed dynamically, then isotopic ordering could be used adjust the number of injections to save time needed to achieve a targeted level of performance. Practically speaking, even if we could vary injection numbers on a vial-to-vial basis, we would need to empirically determine the injection count needed to achieve some threshold of 'memory-freeness' for a given vial-to-vial isotopic difference. The main concern here compared to our current approach is that our standards currently have the same number of injections as our samples. This brings us slightly closer to an ideal Identical Treatment scenario, which we prefer over the idea you propose.
  - The relevant section in the discussion is the same as the previous comment related to memory, at lines 388-405.

**I don't quite understand why they used a particular standard for drift correction. Would it be possible to use one of the other four standards you have in use?**

- This is addressed in the open discussion. We've added some context in the main text (section 2.2).
  - Additional text at lines 156-157.

**Minor points:**

**P5        Table 1 legend: something is missing....by this....**

- Correct, that text got cut off! Thanks!

**P6        Is there a reason why the vial position is different for the standards but not for the MCM QAQC sample (vial position 1 for both the first and last measurements)?**

- The standards are all in separate vials because they begin to evaporate and isotopically fractionate after the septum has been pierced. We are not concerned with the isotopic integrity of the MCM QAQC sample, but rather are interested in whether or not it has highly elevated D17O values due to MCM cartridge failure.

**PXX** **the formatting of d18O and d17O is not consistent, check the complete text.**

- We think we've caught all of these throughout the manuscript.

**P15, l324ff You have measured an unknown sample several times distributed over several series and months. Have you measured the sample water in the same vial? Have you recapped the vial after each measurement sequence? Or have you measured the same water sample by decanting it into individual vials every time you measured it? The procedure is not clear to me it but should be mentioned.**

- We added text in that section to clarify our method. Samples were stored at 4 °C in 4 mL glass vials with polyethylene 'PolyCone' caps wrapped in Parafilm. During measurement, a sample was opened and a 200 μL aliquot was transferred to a measurement vial. Storage vials were then recapped, fresh Parafilm applied, and stored again at 4 °C.
  - Additional text at lines 135-138.

**Fig. 6** **I guess that the values shown depending on the previous sample or standard value measured or is this no issue? The mean absolute error may indicate that it is dependent on the various sample. The accepted value of the current sample is reached either from below or above this value. Therefore, one would have to indicate based on which jump (current to previous sample or standard) these values shown in Fig. 6 are based. The absolute error then depends on this jump. This should be mentioned.**

- Figure 6 is an analysis that converts all the final, fully-calibrated standard values into absolute errors from their true value. The analysis assumes that the behavior of each standard is similar. This appears to be at least roughly true given the similar performance of each standard when summarized separately (Table 3). Of course, a possibly bigger assumption with this figure is that sample performance is similar to standard performance. We strive towards the Identical Treatment principle with our standards to ensure as best as possible that they are a faithful indicator of instrument error and, thus, sample performance.

---

## Author Response (AR2)

Dear Dr. Janssen,

Thanks for the detailed review of our manuscript – your suggestions definitely improve readability and technical clarity. Please see below for our responses to your suggestions. We have included the full text of your review with our responses bulleted.

Best,

Jack Hutchings

Dear authors,

Thank you for your detailed responses to the reviewers' requests, which have been fully addressed. In addition to the specific comments of the reviewers, I have some mostly formal remarks that I believe will improve readability of the manuscript, especially for readers who are not so familiar with the CRDS instrument that you are characterising. Please consider my remarks that are listed below.

I also had difficulties to access the supplementary information and I need to ask the editorial office for these files. Did you make the latest version of these files available ?

- Our supplement is hosted by Open Science Framework under the following URL, as indicated in section 7: osf.io/hgn8k. This link appears to us to be publicly accessible. Please let us know if we are mistaken.

With kind regards,

Christof JANSSEN

1. The title should start with "Optimization of a Picarro" since the work has been undertaking using only a single instrument ....

- Agreed. We have adjusted the title accordingly.

2. The first phrase of the abstract is clumsy to read due to the repetitive use of the word 'measurement'. Please consider to change into 'Until the recent development of comercially available infrared-laser analyzers, the measurement of triple oxygen measurements in water has been restricted to dual-inlet mass spectrometry due to demanding precision requirements.', or similar.

- Agreed. We have replaced the first 'measurements' with 'analyses'.

3. L13. Introduce the acronym CRDS here as cavity ring down spectrometry is mentioned here for the first time.

- Are you sure this is ideal? We have added the acronym CRDS, but we are not sure about the convention of introducing acronyms within the abstract for use in the rest of the manuscript. We note the D17O in the first line of the abstract because D17O is used throughout the abstract, whereas no further use of 'CRDS' is found in the abstract. We are fine either way, of course, but perhaps edit this in or out depending on best convention.

4. L58. I suppose that you mean integrated absorbance values ...

- Yes – we adjusted the text to fit this (L59). We also edited that sentence to hopefully improve clarity.

5. L67. Normal Mode 17O Mode etc. Please use same capitalization and spelling throughout the manuscript. There are occurrences of MCM 17O Long Pulse Mode, 17O Mode, 17O mode, 17O-mode, for example.

Also, please, specify the signification of each mode upon the first mentioning.

- We have added additional text between lines 65 and 85 to help clarify. We also edited text throughout to harmonize usage of the terms with consistent capitalization, hyphenation, etc.

6. L68. Each isotopologues -> each isotopologues

- Fixed.

7. L72. are achieving -> are achieved

- Fixed.

8. L78. the L2140 -> a L2140

- Fixed.

9. Table 1: Use superscript 17 in tablenote b

- Fixed.

10. L125. Please check with the editorial office if v/v is an accepted way of indicating a volume fraction. Use consistent notation, because later in the text v:v is used instead of v/v. At other instances, the mass concentration (in g/L) is given instead of the volume fraction. It would be helpful to use either of the two quantities to specify the alcohol concentration. If for some reason original data are specified differently (eg for different commercial products or data sheet values), please select one of the two quantities to be always indicated (for example, give the mass concentration in addition to v/v). This facilitates the comparison of the different concentrations.

- We converted this first instance (now L128) to mass fraction. We left in the v:v notation in the results (section 3.3) where we also specify the mass fraction in mg / L.

11. L173,L210. Please explain Picarro slang to the reader. Is the data format the only difference between coordinator and HDF format ? Explain differences already here, especially the information that you are making use of in the postprocessing.

- We initially omitted details here as they were previously described by Schauer et al., 2016. However, we added some additional information so that a reader can understand our approach without necessary reference to the Schauer paper.

12. L. 177. What are private data ? The presentation of coordinator data vs high resolution data is somewhat unsatisfactory. What is the temporal resolution of the coordinator data, etc ? It seems that a descritption of what data are provided in the coordinator files is missing in the first place.

- Again, we definitely oversimplified our handling of this. We believe our expanded text in this section should satisfy your questions.

13. L209. For comparison, … The logic of this phrase is not clear. Why has the R script been modified, which script has been modified ? What needs to be compared ? Please rephrase.

- Rephrased to help clarify.

14. L212. Consider citing tidyverse using the recommended source : https://joss.theoj.org/papers/10.21105/joss.01686

- Fixed, thanks.

15. L. 290. Please clarify whether the two delta18O values also depend on independently measured 16O-water peaks.

- Clarified (they use the same 16O-peak) and added some rationale.

16. L. 292. Use of 5.5 and 0.5 is potentially preferrable over 11/2 and 1/2 as the former could eventually be misread as 1 1/2 or 1.5.

- Good catch – these were very poorly formatted ratios of different spectral peaks. We clarify this in the next now without use of the forward-slash.

17. L. 339. 'When using the regression coefficients from Fig. S5 ...': Supplementary material should support/supplement statements made in the manuscript, but the manuscript should be complete in itself and without the supplementary material. This is not the case here, where material from the supplementary material is employed for the reasoning in the main text. Please change the paragraph accordingly.

- Fair… we simply removed this sentence as the estimate (18 months) doesn't have much real meaning given how little effect storage length apparently had on replicate precision.

18. L. 364. 'To compare these...'. Plese consider writing 'To compare the two approaches ...', as 'these' is seems being associated with 'corrections' in the phrase before.

- Fixed, thanks.

19. L. 366. Please replace the word 'improve'. It seems be chosen wrongly as hdf is already your standard approach.

- Edited this sentence to hopefully improve clarity.

20. L. 376. 'avoid needing these corrections'. Please consider removal of the word 'needing'.

- Removed.

21. L. 369. 'In contrast ...' . Why is that ? This is somewhat unexpected and deserves explanation. Is the missing information provided later on in the manuscript ?

- Edited this line for clarity. The "in contrast" is simply meant to indicate that, although we see improvements in short-term precision, we don't find similar improvements in the final accuracy metric (RMSE) of the analysis.

22. L. 477. The 'to' seems to be superficial. Plese delete.

- Correct, thanks.

23. Fig. 6. You should consider a log scale for the abscissa, on which the SD should yield a slope -1/2 line for white (or gaussian) noise dominated errors. On that logarithmic scale it would be interesting to show the 7th and 8th vial (if available).

[Figure]

- See a log10-transformed x-axis of that plot above with minor edits so everything is visible. That being said, our intended purpose of Fig. 6 is to provide the reader with an easily actionable plan as per the required number of analyses to reach a desired level of confidence in an unknown sample. With that intention in mind, we have opted to keep the current format of that figure for ease of useability.

24. L. 490. 'two lasers'. This is the first time that you mention that the instrument is operating with two different lasers. Indeed, it is unclear whether there are two lasers in the instrument and which spectral range they span. As of reading section on the 18O-Laser flag (~ L292) one might also get the impression that there is just one laser that, however, sweeps a wider range and covers two different 18O-containing water absorption lines. Please clarify.

- We briefly note the second laser in the introduction (L50-55), however we expanded the text in the 18O-Laser Flag section to clarify. We left the text you noted initially alone with the thought that the added text in the 18O-Laser Flag section is sufficient.

25. L. 607. '(see Sect 2.3 for details on output files)'. Actually it seems that some of these details are missing in Sect 2.3, see earlier remarks on the coordinator output.

- Our added text from the previous comment should satisfy this, too.

26. L. 610. and elsewhere. Reference is made to h5. It seems this is a short name for HDF v5. If this is so, please harmonize the notation. Otherwise, pease introduce the variable h5 when it is used first.

- You are correct. The extension the Picarro uses is simply h5, but as we noted, it is indeed HDF v5. We have altered text referring to 'h5' to simply be 'HDF'.

27. L. 643. Please consider to delete 'that'.

- Agreed.

28. L. 667-672. This short conclusion refers to three different instances in the supplementary data section. If improved precision is noteworthy in the conclusion, it is preferable that the associated figures (S10, S11) which demonstrate the superior performance appear in the main text.

- We have removed the mention of precision (as this is quite small/negligible) and removed the reference to S11 as we make clear the improvement in d2H at the end of the discussion.

29. Reference section: Please check typography and notation. It seems that sub- and superscripts, as well as delta-symbols in article titles are not always displayed correctly.

- Thanks - we've resolved these typographical issues.